# Stochastic Multi-Armed Bandits with Control Variates

**Arun Verma**
Department of Computer Science
National University of Singapore
arun@comp.nus.edu.sg

**Manjesh K. Hanawal**
Department of IEOR
IIT Bombay, Mumbai, India
mhanawal@iitb.ac.in

## Abstract

This paper studies a new variant of the stochastic multi-armed bandits problem where auxiliary information about the arm rewards is available in the form of control variates. In many applications like queuing and wireless networks, the arm rewards are functions of some exogenous variables. The mean values of these variables are known a priori from historical data and can be used as control variates. Leveraging the theory of control variates, we obtain mean estimates with smaller variance and tighter confidence bounds. We develop an upper confidence bound based algorithm named UCB-CV and characterize the regret bounds in terms of the correlation between rewards and control variates when they follow a multivariate normal distribution. We also extend UCB-CV to other distributions using resampling methods like Jackknifing and Splitting. Experiments on synthetic problem instances validate performance guarantees of the proposed algorithms.

## 1 Introduction

Stochastic Multi-Armed Bandits (MAB) setting has been extensively used to study decision making under uncertainty (Thompson, 1933; Bubeck et al., 2012; Lattimore and Szepesvári, 2020). In the classical setting, it is assumed that the arm rewards are independent of each other. After playing an arm, the learner observers independent and identically distributed reward samples. The exploration versus exploitation trade-off is a fundamental problem in the bandits setting, and a learner can accumulate more rewards if it is balanced well. A better balance can be achieved if the learner can estimate arm's mean rewards with tighter confidence bounds (Auer et al., 2002; Auer and Ortner, 2010; Garivier and Cappé, 2011) and smaller variance (Audibert et al., 2009b; Mukherjee et al., 2018). Any available side or auxiliary information can aid in building tighter confidence bounds. In this paper, we study the improvement in the performance that can be achieved when side information is available in the form of control variates.

Any observable input that is correlated with the random variable of interest can be a Control Variate (CV). When the mean of the CV is known, it becomes useful to reduce the variance of the mean estimator of the random variable. CV method is a popular variance reduction technique to improve the estimates' precision without altering the simulation runs or an increase in the number of samples (Lavenberg et al., 1982). Many stochastic simulations involve inputs generated using known distributions. These inputs can be potential CVs as output could be correlated with them. It has motivated the development of rich theory to analyze the quality of the estimators resulting from CVs (Lavenberg and Welch, 1981; Lavenberg et al., 1982; Nelson, 1990). We adopt these techniques to study the stochastic multi-armed bandits problem when CVs are available for the arms.

In many real-life applications, exogenous variables influence arms' rewards and act as CVs. For example, consider a job aggregating platform that assigns jobs to one of the servers/workers in each slot. The jobs are of variable size, and the time to serve them depends on their size and other

35th Conference on Neural Information Processing Systems (NeurIPS 2021).

unknown extraneous factors. The platform observes the size of each job and knows their mean value from historical data. The random job sizes correlate with service times and can be used as a CV to estimate each server's mean service time. Another example is a wireless system where a transmitter can transmit packets on a channel in each slot. Successful transmission of the packets (quality) depends on random quantities like fading, interference, and channel noise on the selected channel. The transmitter can observe the fading value using pilot signals in each round and know their mean value from past observations. These fading values on a channel can be used as a CV to estimate the channel's quality.

The performance of a MAB algorithm depends on how good is the confidence interval of its mean reward estimators (Auer et al., 2002; Auer and Ortner, 2010; Garivier and Cappé, 2011). The tightness of these intervals depends on the variance of the estimators (Audibert et al., 2009b; Mukherjee et al., 2018). Naturally, estimators with smaller variance for the same number of samples result in better performance. Thus CVs can be leveraged to improve the performance of bandits algorithms with smaller confidence intervals. The reduction depends on how strongly the reward samples and associated CVs are correlated.

Linear methods are widely used for variance reduction using CVs where the new samples are generated taking the weighted sum of reward samples and centered CV samples. Then these new samples are used for the estimation of the mean rewards. The choice of the weight that results in maximum variance reduction depends on the variance of CV and its correlation with reward samples. In practice, both of these quantities are unknown and need to be estimated from the observed samples. However, using the estimated weight makes the new samples highly correlated and makes the analysis of confidence intervals challenging.

For multivariate normal distributions, tight confidence intervals can be constructed that hold with high probability for the mean reward estimators resulting from the linear control variates. Using these results, we develop a MAB algorithm that exploits CVs to improve the regret performance. For general distributions, we discuss resampling methods that result in unbiased estimators with better confidence intervals. Specifically, our contributions are as follows:

- For the case where rewards and CVs are normally distributed, we develop an algorithm named Upper Confidence Bounds with Control Variates (UCB-CV) that uses the estimators based on the linear control variates.

- In Section 4, we show that the regret of UCB-CV is smaller by a factor $(1 - \rho^2)$ compared to the existing algorithms when the rewards and CV have a multivariate normal distribution, where $\rho$ is the correlation coefficient of the reward and control variates.

- In Section 5, we discuss how to adapt UCB-CV for the general distributions using estimators and associated confidence intervals based on jackknifing, splitting, and batching methods.

- We validate the performance of UCB-CV on synthetically generated data in Section 6.

## 1.1 Related Work

Our work uses side or auxiliary information available in the form of CVs to improve the variance of estimates. In the following, we discuss works that incorporate variance estimates and/or side information to improve the performance of multi-armed bandits algorithms.

**Incorporating variance estimates:** Many stochastic multi-armed bandits algorithms like UCB1 (Auer et al., 2002), DMED (Honda and Takemura, 2010), KL-UCB (Garivier and Cappé, 2011), Thompson Sampling (Chapelle and Li, 2011; Agrawal and Goyal, 2012; Kaufmann et al., 2012; Agrawal and Goyal, 2013), assume that the rewards are bounded on some interval $[0, b]$ and define index of each arm based on its estimates of the mean rewards and ignore the variance estimates. The regret of these algorithms have optimal logarithmic bound in the number of rounds $(T)$, and grows quadratic in $b$, i.e., $\mathcal{O}((b^2/\Delta) \log T)$, where $\Delta$ is the smallest sub-optimality gap. UCB1-NORMAL by (Auer et al., 2002) uses the variance estimates in the index of the arms and when rewards are Gaussian with variance $\sigma^2$, to achieve regret is of order $\mathcal{O}((\sigma^2/\Delta) \log T)$. This bound is even better than that of UCB1 when the variance is smaller than $b^2$ even though it has considered unbounded support. Further, the authors experimentally demonstrate that the UCB-Tuned algorithm (Auer et al., 2002), which uses variance estimates in arm indices, significantly improves regret performance over UCB1 when rewards are bounded. Extending the idea of using 'variance aware' indices, UCBV

(Audibert et al., 2009b) algorithm achieves regret of order $\mathcal{O}((\sigma^2/\Delta + 2b) \log T)$ thus confirming that making algorithms variance aware is beneficial. EUCBV (Mukherjee et al., 2018) improves the performance of UCBV using an arm elimination strategy like used in UCB-Improved (Auer and Ortner, 2010).

**Incorporating side information:** In the applications like advertising and web search, whether or not users like a displayed item could depend on their profile. Users' profile is often available to the platform, which could then be used as side information. Contextual bandits (Dani et al., 2008; Rusmevichientong and Tsitsiklis, 2010; Li et al., 2010; Filippi et al., 2010; Abbasi-Yadkori et al., 2011; Chu et al., 2011; Li et al., 2017; Jun et al., 2017; Zhang et al., 2016) and MAB with covariates (Perchet and Rigollet, 2013) exploit such side information to learn the optimal arm. They assume a correlation structure (linearity, GLM, etc.) between mean reward and context where mean reward varies with side-information (context).

In contrast to contextual bandits, mean rewards do not vary in our unstructured setup with side information available in the form of CVs. We do not use CVs directly to decide which arm to select in each round but use them only to get better estimates of the mean rewards by exploiting their correlation with rewards. Our motivating examples from jobs scheduling and communication networks capture these unstructured bandits where CVs provide information about the rewards but do not alter their mean values.

Our confidence bounds for mean reward estimators are based on the variance of estimators rather than on the variance of the reward sample, as was the case in UCBV (Audibert et al., 2009a) and EUCBV (Mukherjee et al., 2018). The samples that we use to estimate mean rewards are not independent and identically distributed. Hence, we can not use the standard Bernstein inequality to get confidence bounds on mean and variance estimates.

CVs are used extensively for variance reduction (James, 1985; Botev and Ridder, 2014; Chen and Ghahramani, 2016; Kreutzer et al., 2017; Vlassis et al., 2019) in the Monte-Carlo simulation of complex systems (Lavenberg and Welch, 1981; Lavenberg et al., 1982; Nelson, 1989, 1990). To the best of our knowledge, our work is the first to analyze the performance of stochastic bandits algorithms with control variates.

## 2   Control Variate for Variance Reduction

Let $\mu$ be the unknown quantity that needs to be estimated, and $X$ be an unbiased estimator of $\mu$, i.e., $\mathbb{E}[X] = \mu$. A random variable $W$ with known expectation $(\omega)$ is a CV if it is correlated with $X$. Linear control methods use errors in estimates of known random variables to reduce error in the estimation of an unknown random variable. For any choice of a coefficient $\beta$, define a new estimator as $\bar{X} = X + \beta(\omega - W)$. It is straightforward to verify that its variance is given by

$$\text{Var}(\bar{X}) = \text{Var}(X) + \beta^2 \text{Var}(W) - 2\beta \text{Cov}(X, W).$$

and it is minimized by setting $\beta$ to

$$\beta^\star = \frac{\text{Cov}(X, W)}{\text{Var}(W)}.$$

The minimum value of the variance is given by $\text{Var}(\bar{X}) = (1 - \rho^2)\text{Var}(X)$, where $\rho$ is the correlation coefficient of $X$ and $W$. Larger the correlation, the greater the variance reduction achieved by the CV. In practice, the values of $\text{Cov}(X, W)$ and $\text{Var}(W)$ are unknown and need to be estimated to compute the best approximation for $\beta^\star$.

## 3   Problem Setting

We consider a stochastic multi-armed bandits problem with $K$ arms. The set of arms is represented by $[K] \doteq \{1, 2, \ldots, K\}$. In round $t$, the environment generates a vector $(\{X_{t,i}\}_{i \in [K]}, \{W_{t,i}\}_{i \in [K]})$. The random variable $X_{t,i}$ denotes the reward of arm $i$ in round $t$, and form an Independent and Identically Distributed (IID) sequence drawn from an unknown but fixed distribution with mean $\mu_i$ and variance $\sigma_i^2$. The random variable $W_{t,i}$ is the Control Variate (CV) associated with the reward of arm $i$ in round $t$. These random variables are drawn from an unknown but fixed distribution with mean $\omega_i$ and variance $\sigma_{w,i}^2$ and form an IID sequence. The learner knows the values $\{\omega_i\}_{i \in [K]}$ but

not the $\{\sigma_{w,i}^2\}_{i\in[K]}$. The correlation coefficient between rewards and associated control variates of an arm $i$ is denoted by $\rho_i$.

In our setup, the learner observes the reward and associated CVs from the selected arm. We refer to this new variant of the multi-armed bandits problem as Multi-Armed Bandits with Control Variates (MAB-CV). The parameter vectors $\boldsymbol{\mu^\sigma} = \{\mu_i, \sigma_i^2\}_{i\in[K]}$, $\boldsymbol{\omega^\sigma} = \{\omega_i, \sigma_{w,i}^2\}_{i\in[K]}$, and $\boldsymbol{\rho} = \{\rho_i\}_{i\in[K]}$ identify an instance of MAB-CV problem, which we denote henceforth using $P = (\boldsymbol{\mu^\sigma}, \boldsymbol{\omega^\sigma}, \boldsymbol{\rho})$. The collections of all MAB-CV problems are denoted by $\mathcal{P}$. For a problem instance $P \in \mathcal{P}$ with mean reward vector $\boldsymbol{\mu}$, we denote the optimal arm as $i^\star = \underset{i\in[K]}{\arg\max}\, \mu_i$.

---

**MAB-CV problem** with instance $(\boldsymbol{\mu^\sigma}, \boldsymbol{\omega^\sigma}, \boldsymbol{\rho})$

For round $t$:

1. **Environment** generates a vector $\boldsymbol{X_t} = (X_{t,1}, \ldots, X_{t,K}) \in \mathbb{R}^K$ and $\boldsymbol{W_t} = (W_{t,1}, \ldots, W_{t,K}) \in \mathbb{R}^K$, where $\mathbb{E}[X_{t,i}] = \mu_i$, $\mathrm{Var}(X_{t,i}) = \sigma_i^2$, $\mathbb{E}[W_{t,i}] = \omega_i$, $\mathrm{Var}(W_{t,i}) = \sigma_{w,i}^2$, and the sequence $(X_{t,i})_{t\geq 1}$ and $(W_{t,i})_{t\geq 1}$ are IID for all $i \in [K]$.

2. **Learner** selects an arm $I_t \in [K]$ based on past observation of rewards and CVs samples from the arms till round $t - 1$.

3. **Feedback and Regret:** The learner observes a random reward $X_{t,I_t}$ and associated CV $W_{t,I_t}$, and then incurs penalty $(\mu_{i^\star} - \mu_{I_t})$.

---

Let $I_t$ denote the arm selected by the learner in round $t$ based on the observation of past reward and control variate samples. The interaction between the environment and a learner is given in MAB-CV problem. We aim to learn policies that accumulate maximum reward and measure its performance by comparing its cumulative reward with that of an Oracle that plays the optimal arm in each round. Specifically, we measure the performance in terms of regret defined as follows:

$$\mathfrak{R}_T = T\mu_{i^\star} - \mathbb{E}\left[\sum_{t=1}^{T} X_{t,I_t}\right]. \tag{1}$$

Note that maximizing the mean cumulative reward of a policy is the same as minimizing the policy's regret. A good policy should have sub-linear expected regret, i.e., $\mathfrak{R}_T/T \to 0$ as $T \to \infty$.

Our goal is to learn a policy that is sub-linear with small regret. To this effect, we use CVs to estimate mean rewards with sharper confidence bounds so that the learner can have a better exploration-exploitation trade-off and start playing the optimal arm more frequently early.

## 4 Arms with Normally Distributed Rewards and Control Variates

We first focus on the case where the rewards and associated CVs of arms have a multivariate normal distribution. We discuss the general distributions in the next section. To bring out the main ideas of the algorithm, we first consider the case where each arm is associated with only one CV. Motivated by the linear CV technique discussed in the previous section, we consider a new sample for an arm $i$ in round $t$ as follows:

$$\bar{X}_{t,i} = X_{t,i} + \beta_i^*(\omega_i - W_{t,i}), \tag{2}$$

where $X_{t,i}$ is the $t^{\text{th}}$ reward, $W_{t,i}$ is the $t^{\text{th}}$ associated control variate with an arm $i$, $\omega_i = \mathbb{E}[W_{t,i}]$, and $\beta_i^* = \mathrm{Cov}(X_{t,i}, W_{t,i})/\mathrm{Var}(W_{t,i})$. Using $s$ such samples, the mean reward for an arm $i$ is defined as $\hat{\mu}_{s,i}^c = (\sum_{r=1}^{s} \bar{X}_{r,i})/s$. Let $\hat{\mu}_{s,i} = \frac{1}{s}\sum_{r=1}^{s} X_{r,i}$ and $\hat{\omega}_{s,i} = \frac{1}{s}\sum_{r=1}^{s} W_{r,i}$ denote the sample mean of reward and CVs of an arm $i$ from $s$ samples. Then $\hat{\mu}_{s,i}^c$ can be written as follows:

$$\hat{\mu}_{s,i}^c = \hat{\mu}_{s,i} + \beta_{s,i}^*(\omega_i - \hat{\omega}_{s,i}). \tag{3}$$

Since the value of $\mathrm{Cov}(X_i, W_i)$ and $\mathrm{Var}(W_i)$ are unknown, the optimal coefficient $\beta_{s,i}^*$ need to be estimated. After having $s$ samples, it is estimated as below and used in Eq. (3).

$$\hat{\beta}_{s,i}^* = \frac{\sum_{r=1}^{s}(X_{r,i} - \hat{\mu}_{s,i})(W_{r,i} - \omega_i)}{\sum_{r=1}^{s}(W_{r,i} - \omega_i)^2}. \tag{4}$$

When the variance of CV is known, it can be directly used to estimate $\beta_i^*$ by replacing estimated variance by actual variance of CV in the denominator of Eq. (4), i.e., replacing $\sum_{r=1}^{s}(W_{r,i} - \omega_i)^2/(s-1)$ by $\sigma_{w,i}^2$. After observing reward and associated control variates for arm $i$, the value of $\hat{\beta}_i^\star$ is re-estimated. The value of $\hat{\beta}_i^\star$ depends on all observed rewards and associated control variates from arm $i$. Since all $\bar{X}_{.,i}$ uses same $\hat{\beta}_i^\star$, this leads to correlation between the $\bar{X}_{.,i}$ observations.

Let $\text{Var}(\bar{X}_i) = \sigma_{c,i}^2$ denote the variance of the linear CV samples and $\nu_{s,i} = \text{Var}(\hat{\mu}_{s,i}^c)$ denote the variance the estimator using these samples. Since the mean reward estimator ($\hat{\mu}_{s,i}^c$) is computed using the correlated samples, we cannot use Bernstein inequality to get confidence bounds on mean reward estimates. However, when the rewards and CVs follow a multivariate normal distribution, the following results tell how to obtain an unbiased variance estimator and confidence interval for $\hat{\mu}_{s,i}^c$.

**Lemma 1.** *Let the reward and control variate of each arm have a multivariate normal distribution. After observing $s$ samples of reward and control variate from arm $i$, define $\hat{\nu}_{s,i} = \frac{Z_{s,i}\hat{\sigma}_{c,i}^2(s)}{s}$, where*

$$Z_{s,i} = \left(1 - \frac{\left(\sum_{r=1}^{s}(W_{r,i}-\omega_i)\right)^2}{s\sum_{r=1}^{s}(W_{r,i}-\omega_i)^2}\right)^{-1} \quad \text{and} \quad \hat{\sigma}_{c,i}^2(s) = \frac{1}{s-2}\sum_{r=1}^{s}(\bar{X}_{r,i}-\hat{\mu}_{s,i}^c)^2,$$

*then $\hat{\nu}_{s,i}$ is an unbiased variance estimator of $\hat{\mu}_{s,i}^c$, i.e., $\mathbb{E}[\hat{\nu}_{s,i}] = \text{Var}(\hat{\mu}_{s,i}^c)$.*

The confidence interval for reward estimator is given by our next result.

**Lemma 2.** *Let the conditions in Lemma 1 hold and $s$ be the number of reward and associated control variate samples from arm $i$ in round $t$. Then*

$$\mathbb{P}\left\{|\hat{\mu}_{s,i}^c - \mu_i| \geq V_{t,s}^{(\alpha)}\sqrt{\hat{\nu}_{s,i}}\right\} \leq 2/t^\alpha,$$

*where $V_{t,s}^{(\alpha)}$ denote $100(1-1/t^\alpha)^{th}$ percentile value of the $t-$distribution with $s-2$ degrees of freedom and $\hat{\nu}_{s,i}$ is an unbiased estimator for variance of $\hat{\mu}_{s,i}^c$.*

We prove Lemma 1 and Lemma 2 using regression theory and control theory results (Nelson, 1990). The detailed proofs are given in Appendix B. The proof uses the fact that the arm's rewards and CVs have a multivariate normal distribution. Therefore, we can use $t$-distribution for designing confidence intervals of mean reward estimators. Analogous to the Hoeffding inequality, this result shows that the probability of the estimate obtained by samples $\{\bar{X}_{t,i}\}$ deviating from the true mean of arm $i$ decays fast. Further, the deviation factor $V_{t,s}^{(\alpha)}\sqrt{\hat{\nu}_{s,i}}$ depends on the variance of the estimator ($\hat{\mu}_{s,i}^c$), which guarantees sharper confidence intervals. As we will see later in Lemma 3, these confidence terms are smaller by a factor $(1-\rho_i^2)$ compared to the case when no CVs are used. Equipped with these results, we next develop a Upper Confidence Bound (UCB) based algorithm for the MAB-CV problem.

### 4.1 Algorithm: UCB-CV

Let $N_i(t)$ be the number of times the arm $i$ is selected by a learner until round $t$ and $\hat{\nu}_{N_i(t),i}$ be the unbiased sample variance of mean reward estimator ($\hat{\mu}_{N_i(t),i}^c$) for arm $i$. Motivated from Lemma 2, we define optimistic upper bound for mean reward estimate of arm $i$ as follows:

$$\text{UCB}_{t,i} = \hat{\mu}_{N_i(t),i}^c + V_{t,N_i(t)}^{(\alpha)}\sqrt{\hat{\nu}_{N_i(t),i}}. \tag{5}$$

Using above values as UCB indices of arms, we develop an algorithm named UCB-CV for the MAB-CV problem. The algorithm works as follows: It takes the number of arms $K$, a constant $Q = 3$ (number of CV per arm + 2), and $\alpha > 1$ (trades-off between exploration and exploitation) as input. Each arm is played $Q$ times to ensure the sample variance for observations ($\hat{\sigma}_{c,i}^2(s)$) can be computed (see Lemma 1).

In round $t$, UCB-CV computes the upper confidence bound of each arm's mean reward estimate using Eq. (5) and selects the arm having highest upper confidence bound value. We denote the selected arm by $I_t$. After playing arm $I_t$, the reward $X_{t,I_t}$ and associated control variate $W_{t,I_t}$ are observed. After that, the value of $N_{I_t}(t)$ is updated and $\hat{\beta}_{N_{I_t}(t),I_t}^*$, $\hat{\mu}_{N_{I_t}(t),I_t}^c$ and $\hat{\nu}_{t,N_{I_t}(t)}$ are re-estimated. The same process is repeated for the subsequent rounds.

---

**UCB-CV** UCB based Algorithm for MAB-CV problem

---

1: **Input:** $K$, $Q$, $\alpha > 1$
2: Play each arm $i \in [K]$ $Q$ times
3: **for** $t = QK + 1, QK + 2, \ldots,$ **do**
4:     $\forall i \in [K]$ : compute $\text{UCB}_{t-1,i}$ as given in Eq. (5)
5:     Select $I_t = \underset{i \in [K]}{\arg\max}\ \text{UCB}_{t-1,i}$
6:     Play arm $I_t$ and observe $X_{t,I_t}$ and associated control variates $W_{t,I_t}$. Increment the value of $N_{I_t}(t)$ by one and re-estimate $\hat{\beta}^*_{N_{I_t}(t),I_t}$, $\hat{\mu}^c_{N_{I_t}(t),I_t}$ and $\hat{\nu}_{t,N_{I_t}(t)}$
7: **end for**

---

## 4.2 Estimator with Multiple Control Variates

In some applications, it could be possible that each arm is associated with multiple CVs. We denote the number of CVs with each arm as $q$. Let $W_{t,i,j}$ be the $j^{\text{th}}$ control variate of arm $i$ that is observed in round $t$, . Then the unbiased mean reward estimator for arm $i$ with associated CVs is given by

$$\hat{\mu}^c_{s,i,q} = \hat{\mu}_{s,i} + \hat{\boldsymbol{\beta}}^{*\top}_i (\boldsymbol{\omega}_i - \hat{\boldsymbol{\omega}}_{s,i}),$$

where $\hat{\boldsymbol{\beta}}^*_i = \left(\hat{\beta}_{i,1}, \ldots, \hat{\beta}_{i,q}\right)^\top$, $\boldsymbol{\omega}_i = (\omega_{i,1}, \ldots, \omega_{i,q})^\top$, and $\hat{\boldsymbol{\omega}}_{s,i} = (\hat{\omega}_{s,i,1}, \ldots, \hat{\omega}_{s,i,q})^\top$.

Let $s$ be the number of rewards and associated CVs samples for arm $i$, $\boldsymbol{W}_i$ be the $s \times q$ matrix whose $r^{\text{th}}$ row is $(W_{r,i,1}, W_{r,i,2}, \ldots, W_{r,i,q})$, and $\boldsymbol{X}_i = (X_{1,i}, \ldots, X_{s,i})^\top$. By extending the arguments used in Eq. (4) to $q$ control variates, the estimated coefficient vector is given by

$$\hat{\boldsymbol{\beta}}^*_i = (\boldsymbol{W}_i^\top \boldsymbol{W}_i - s\hat{\boldsymbol{\omega}}_{s,i}\hat{\boldsymbol{\omega}}_{s,i}^\top)^{-1}(\boldsymbol{W}_i^\top \boldsymbol{X}_i - s\hat{\boldsymbol{\omega}}_i\ \hat{\mu}_{s,i}).$$

We can generalize Lemma 1 and Lemma 2 for the MAB-CV problems with $q$ control variates and then use UCB-CV with $Q = q + 2$ and appropriate optimistic upper bound for multiple control variate case. More details can be found in Appendix C.

## 4.3 Analysis of UCB-CV

In our next result, we describe the property of our estimator that uses control variates. This result is derived from the standard results of control variates theory (Nelson, 1990).

**Lemma 3.** *Let reward and control variates have a multivariate normal distribution and $q$ be the number of control variates. Then after having $s$ observations of rewards and associated control variates from arm $i$,*

$$\mathbb{E}\left[\hat{\mu}^c_{s,i,q}\right] = \mu_i, \text{ and}$$

$$Var(\hat{\mu}^c_{s,i,q}) = \frac{s-2}{s-q-2}(1-\rho_i^2)Var(\hat{\mu}_{s,i}),$$

*where $\rho_i^2 = \sigma_{X_i \boldsymbol{W}_i} \Sigma^{-1}_{\boldsymbol{W}_i \boldsymbol{W}_i} \sigma^\top_{X_i \boldsymbol{W}_i} / \sigma_i^2$ is the square of the multiple correlation coefficient, $\sigma_i^2 = Var(X_i)$, and $\sigma_{X_i \boldsymbol{W}_i} = (Cov(X_i, W_{i,1}), \ldots, Cov(X_i, W_{i,q}))$.*

The following is our main result which gives the regret upper bound of UCB-CV. The detailed analysis is given in Appendix D.

**Theorem 1.** *Let the conditions in Lemma 3 hold, $\alpha = 2$, $\Delta_i = \mu_{i^\star} - \mu_i$ be the sub-optimality gap for arm $i \in [K]$, and $N_i(T)$ be the number of times sub-optimal arm $i$ selected in $T$ rounds. Let $C_{T,i,q} = \mathbb{E}\left[\frac{N_i(T)-2}{N_i(T)-q-2}\left(\frac{V^{(2)}_{T,N_i(T),q}}{V^{(2)}_{T,T,q}}\right)^2\right]$ for all $i$. Then the regret of UCB-CV in $T$ rounds is upper bounded by*

$$\mathfrak{R}_T \leq \sum_{i \neq i^\star} \left(\frac{4(V^{(2)}_{T,T,q})^2 C_{T,i,q}(1-\rho_i^2)\sigma_i^2}{\Delta_i} + \frac{\Delta_i \pi^2}{3} + \Delta_i\right).$$

Note that $\alpha$ is a hyper-parameter to trade-off between exploration and exploitation. UCB-CV works as long as $\alpha > 1$ and the effect of $\alpha$ on the regret bound is through the term $\sum_{i=1}^{\infty} \frac{1}{t^{\alpha}}$. This term has a nice closed form value of $\pi^2/6$ for $\alpha = 2$. Hence, we state the results with $\alpha = 2$ and use it in the experiments as well.

**Remark 1.** *Unfortunately, we cannot directly compare our bound with that of UCB based stochastic MAB algorithms like UCB1-NORMAL and UCBV as $V_{T,T,q}^{(2)}$ do not have closed-form expression. $V_{T,T,q}^{(2)}$ denotes $100(1 - 1/T^2)^{th}$ percentile of $t$-distribution with $T - q - 1$ degrees of freedom.*

**Remark 2.** *As the degrees of freedom decreases in $q$, $V_{T,T,q}^{(2)}$ increases in $q$ (see Fig. 1a for small value of $T$ and Fig. 1b for large value of $T$). However, this does not imply that the overall regret increases in $q$ as $\rho_i^2$ also increases in $q$. Hence dependency of regret on $q$ is delicate. It is also observed in Nelson (1989) that having more CVs does not mean better variance reduction. One can empirically verify that $(V_{T,T,q}^{(2)})^2$ is upper bounded by $3.726 \log T$. This upper bound holds for all $q$ as long as $T \geq q + \max(32, 0.7q)$.*

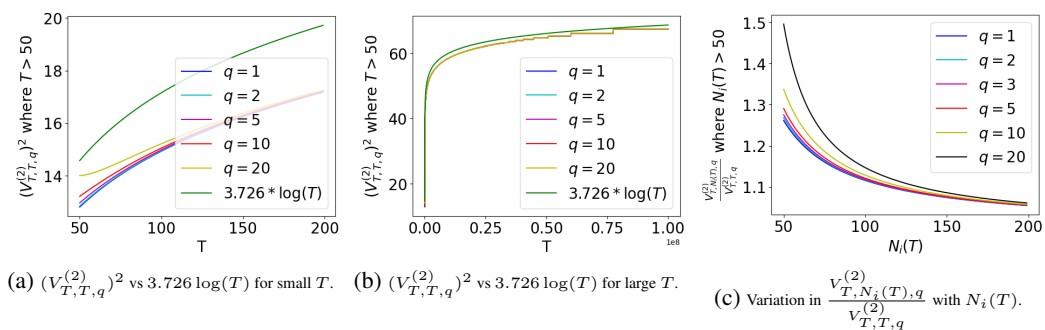

(a) $(V_{T,T,q}^{(2)})^2$ vs $3.726 \log(T)$ for small $T$.    (b) $(V_{T,T,q}^{(2)})^2$ vs $3.726 \log(T)$ for large $T$.    (c) Variation in $\frac{V_{T,N_i(T),q}^{(2)}}{V_{T,T,q}^{(2)}}$ with $N_i(T)$.

Figure 1: Properties of $(V_{T,T,q}^{(2)})^2$ and $V_{T,N_i(T),q}^{(2)}/V_{T,T,q}^{(2)}$.

**Remark 3.** *It is hard to bound $C_{T,i,q}$ as $V_{T,\cdot,q}^{(2)}$ does not have a explicit form. However, $C_{T,i,q}$ tends to 1 as $T \to \infty$. Indeed, from lower bound argument, we know that $N_i(T) \to \infty$ as $T \to \infty$ and hence $\frac{N_i(T)-2}{N_i(T)-q-2} \searrow 1$ and $\frac{V_{T,N_i(T),q}^{(2)}}{V_{T,T,q}^{(2)}} \searrow 1$ (confidence interval shrink with more samples) as shown in Fig. 1c. However, under some assumptions, we can obtain a bound empirically. Specifically, if $q \leq 20$ and $N_i(T) > 50$, then $\frac{N_i(T)-2}{N_i(T)-q-2} \leq 1.72$ and from $t$-distribution tables, we get $\frac{V_{T,N_i(T),q}^{(2)}}{V_{T,T,q}^{(2)}} \leq 1.5$ (see Fig. 1c), hence the bound $C_{T,i,q} \leq 4$. We can guarantee $N_i(T) > 50$ by playing each arm 50 times at the beginning. Therefore, we can obtain one possible explicit bound in Lemma (6) by replacing $C_{T,i,q}$ with 4. However, for a general case, an explicit bound in Lemma 8 is hard.*

## 5 General Distribution

We now consider the general case with no distributional assumptions on reward and associated CVs of arms. The samples $\bar{X}_{t,i,q}$ resulting from the linear combination of rewards and CVs are dependent but need not be normally distributed. Hence $\hat{\mu}_{s,i,q}^c$ need not remain an unbiased estimator. Therefore, we cannot use the $t$-distribution properties to obtain confidence intervals. However, we can use resampling methods such as jackknifing, splitting, and batching to reduce the bias of the estimators and develop confidence intervals that hold approximately or asymptotically. These confidence intervals can then be used to defined indices for the arms. Below we briefly discuss these methods.

### 5.1 Jackknifing

In statistics, jackknifing is a well-known re-sampling technique for variance and bias estimation (Efron, 1982; Efron and Gong, 1983). We start with a classical presentation of jackknifing for CVs

(Lavenberg et al., 1982). Let $\hat{\mu}_{s,i,q}^{c,-j}$ be the estimate of the mean reward that is computed without sample $\bar{X}_j$. The $Y_{j,i,q}^{\text{J}} = s\hat{\mu}_{s,i,q}^{c} - (s-1)\hat{\mu}_{s,i,q}^{c,-j}$, which is sometimes called the $j^{\text{th}}$ pseudo-value. Using pseudo-values mean reward estimation given as $\hat{\mu}_{s,i,q}^{c,\text{J}} = \frac{1}{s}\sum_{j=1}^{s} Y_{j,i,q}^{\text{J}}$ and its sample variance is $\hat{\nu}_{s,i,q}^{\text{J}} = (s(s-1))^{-1}\sum_{j=1}^{s}(Y_{j,i,q}^{\text{J}} - \hat{\mu}_{s,i,q}^{c,\text{J}})^2$. $\hat{\mu}_{s,i,q}^{c,\text{J}}$ is shown to be unbiased (Nelson, 1989)[Thm. 7] when reward variance is bounded. Further, $\hat{\mu}_{s,i,q}^{c,\text{J}} \pm t_{\alpha/2}(n-1)\hat{\nu}_{s,i,q}^{\text{J}}$ is a asymptotically valid confidence interval (Avramidis et al., 1991) and holds approximately for finite number of samples.

## 5.2 Splitting

Splitting is a technique to split the correlated observations into two or more groups, compute an estimate of $\boldsymbol{\beta}_i^*$ from each group, then exchange the estimates among the groups. Nelson (1990) considers an extreme form of splitting, where he splits $s$ observations into $s$ groups. The $j^{\text{th}}$ observation is given by

$$Y_{j,i,q}^{\text{S}} = X_{j,i} + \hat{\boldsymbol{\beta}}_i^{*-j}(\boldsymbol{\omega} - \boldsymbol{W}_{j,i}), \quad j \in [s],$$

where $\hat{\boldsymbol{\beta}}_i^{*-j}$ is estimated without $j^{\text{th}}$ observation and $\boldsymbol{W}_{j,i} = (W_{j,i,1}, \ldots, W_{j,i,q})$ is the vector of CVs with $q$ elements associated with reward $X_{j,i}$. The point estimator for splitting method is $\hat{\mu}_{s,i,q}^{c,\text{S}} = \frac{1}{s}\sum_{j=1}^{s} Y_{j,i,q}^{\text{S}}$ and its sample variance is $\hat{\nu}_{s,i,q}^{\text{S}} = (s(s-1))^{-1}\sum_{j=1}^{s}(Y_{j,i,q}^{\text{S}} - \hat{\mu}_{s,i,q}^{c,\text{S}})^2$. Then $\hat{\mu}_{s,i,q}^{c,\text{S}} \pm t_{\alpha/2}(n-1)\hat{\nu}_{s,i,q}^{\text{S}}$ gives an approximate confidence interval (Nelson, 1990).

Below we define UCB index for these methods. Let $\hat{\nu}_{N_i(t),i,q}^{\text{G}}$ be the sample variance of mean reward estimator $(\hat{\mu}_{s,i,q}^{c,\text{G}})$ for arm $i$. We define the optimistic upper bound for mean reward for $G \in \{J, S\}$ as follows:

$$\text{UCB}_{t,i,q}^{\text{G}} = \hat{\mu}_{N_i(t),i,q}^{c,\text{G}} + V_{t,N_i(t),q}^{\text{G},\alpha}\sqrt{\hat{\nu}_{N_i(t),i,q}^{\text{G}}}. \tag{6}$$

where $V_{t,s,q}^{\text{G},\alpha}$ is the $100(1 - 1/t^{\alpha})^{\text{th}}$ percentile value of the $t-$distribution with $s - 1$ degrees of freedom. Since the optimistic upper bounds defined in Eq. (6) are valid only asymptotically (Nelson, 1990; Avramidis et al., 1991), it cannot be used for any finite time regret guarantee. However the above UCB indices can be used in UCB-CV to get an heuristic algorithm for the general case. We experimentally validate its performance in the next section.

## 6 Experiments

We empirically evaluate the performance of UCB-CV by comparing it with UCB-CV with UCB1 (Auer et al., 2002), UCB-V (Audibert et al., 2009b), Thompson Sampling (Agrawal and Goyal, 2013), and EUCBV (Mukherjee et al., 2018) on different synthetically generated problem instances. For all the instance we use we use $K = 10$, $q = 1$, and $\alpha = 2$. All the experiments are repeated 100 times and cumulative regret with a 95% confidence interval (the vertical line on each curve shows the confidence interval) are shown. Details of each instance are as follows:

*Instance 1:* The reward and associated CV of this instance have a multivariate normal distribution. The reward of each arm has two components. We treated one of the components as CV. In round $t$, the reward of arm $i$ is given as follows:

$$X_{t,i} = V_{t,i} + W_{t,i},$$

where $V_{t,i} \sim \mathcal{N}(\mu_{v,i}, \sigma_{v,i}^2)$ and $W_{t,i} \sim \mathcal{N}(\mu_{w,i}, \sigma_{w,i}^2)$. Therefore, $X_{t,i} \sim \mathcal{N}(\mu_{v,i}+\mu_{w,i}, \sigma_{v,i}^2+\sigma_{w,i}^2)$. We treat $W_i$ as CV of $X_i$. It can be easily shown that the correlation coefficient of $X_i$ and $W_i$ is $\rho_i = \sqrt{\sigma_{w,i}^2/(\sigma_{v,i}^2 + \sigma_{w,i}^2)}$. For each $i$, we set $\mu_{v,i} = 0.6 - (i-1)*0.05$, $\mu_{w,i} = 0.3$, for arm $i \in [K]$. The value of $\sigma_{v,i}^2 = 0.1$ and $\sigma_{w,i}^2 = 0.1$ for all arms. Note that the problem instance has the same CV for all arms, but all the CVs observations for each arm are maintained separately.

*Instance 2:* It is the same as the Instance 1 except each arm has a different CV associated with its reward. The mean value of the CV associated with arm $i$ is set as $\mu_{w,i} = 0.8 - (i-1)*0.05$.

*Instance 3:* It is the same as the Instance 2 except the samples of reward and associated control variate are generated from Gamma distribution, where the value of scale is set to 1.

*Instance 4:* It is the same as the Instance 2 except the samples of reward and associated control variate are generated from Log-normal distribution with the values of $\sigma_{v,i}^2 = 1$ and $\sigma_{w,i}^2 = 1$ for all arms.

**Comparing regret of UCB-CV with existing algorithms** The regret of different algorithms for Instances 1 and 2 are shown in Figures Fig. 2a and Fig. 2b, respectively. As see UCB-CV outperforms all the algorithms. We observe that Thomson Sampling has a large regret for normally distributed reward and CVs. Hence, we have not added regret of Thomson Sampling. Note that UCB-CV does not require a bounded support assumption, which is needed in all other algorithms. The regret of the EUCBV algorithm is closest to UCB-CV, but EUCBV can stick with the sub-optimal arm as it uses an arm elimination-based strategy, which has a small probability of eliminating the optimal arm.

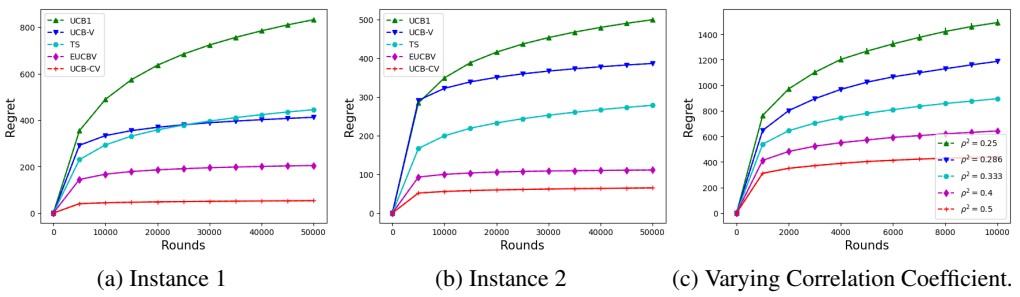

(a) Instance 1       (b) Instance 2       (c) Varying Correlation Coefficient.

Figure 2: Regret comparison of different multi-armed bandits algorithms with UCB-CV.

**Regret of UCB-CV vs Correlation Coefficient** Theorem 1 shows that UCB-CV have better regret bounds when correlation between arm rewards and CVs is higher. To validate this we derived problem instances having different correlation coefficients as follows:

*Instance 5:* Similar to the Instance 1, the reward and associated CVs have a multivariate normal distribution with rewards expressed as sum of two components. We set $\mu_{v,i} = 6.0 - (i-1) * 0.5$ and $\mu_{w,i} = 4.0$ for arm $i \in [K]$. The value of $\sigma_{v,i}^2 = 1.0$ and $\sigma_{w,i}^2 = 1.0$ for all arms. The problem instance has common CV for all arms but all the observations are maintained for each arm separately. As the correlation coefficient of $X_i$ and $W_i$ is $\rho_i = \sqrt{\sigma_{w,i}^2/(\sigma_{v,i}^2 + \sigma_{w,i}^2)}$, we varied $\sigma_{v,i}^2$ over the values $\{1.0, 1.5., 2.0, 2.5, 3.0\}$ to obtain problem instances with different correlation coefficient. Increasing $\sigma_{v,i}^2$ reduces the correlation coefficient of $X_i$ and $W_i$. We have plotted the regret of UCB-CV for different problem instances. As expected, we observe that the regret decreases as the correlation coefficient of reward and its associated control variates increases as shown in Fig. 2c.

**Performance of Jackknifing, Splitting, and Batching Methods** We compare the regret of UCB-CV with Jackknifing, Splitting and Batching (see Appendix E for more details) methods. The regret of the batching method is worse for all the instances. Whereas, the jackknifing and splitting are performing well for heavy-tailed distributions (Instance 3) and UCB-CV performs better for normal distribution (Instance 2) and Log-normal distribution (Instance 4) as shown in Fig. 3.

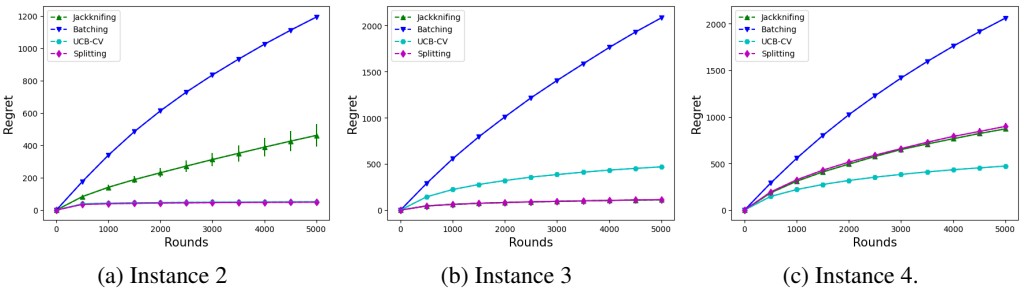

(a) Instance 2       (b) Instance 3       (c) Instance 4.

Figure 3: Comparing regret of UCB-CV with Jackknifing, Splitting, and Batching method based.

**Regret of UCB-CV vs Estimated Mean of CV**   Our work has demonstrated the applicability of CVs to bandit theory. To establish the gains analytically, we have to assume that mean of the CVs is known. However, it is not unreasonable as the mean of CVs can be constructed such that its mean value is known (see Eq. (7) and Eq. (8) of Kreutzer et al. (2017)). If the mean of CV is unknown, we estimate it from the samples, but using the estimated/approximated means can deteriorate the performance of the proposed algorithm.s Though the theory of control variate is well developed for known mean, and only empirical studies are available to the case when the CV means are known approximately (Schmeiser et al., 2001; Pasupathy et al., 2012).

To know the effect of approximation error ($\varepsilon$) in the mean estimation of CVs, we run an experiment with Instance 1, Instance 2, and Instance 5. We assume that the approximated mean of CVs is given by $\omega_i + \varepsilon$. We observe that the regret of UCB-CV increases with an increase in approximation error. UCB-CV can even start performing poorly than the existing state-of-the-art algorithm for significant large approximation errors as shown in Fig. 4. Since the maximum and minimum reward gap can be more than one for the Instance 5, we must multiply confidence intervals of UCB1, UCB-V, and EUCBV with the appropriate factor (used 6 for the experiment) for a fair comparison. We also observe that Thompson Sampling has almost linear regret for the Instance 5. We believe that it is due to the high variance of arms' rewards that leads to overlapping of rewards distributions.

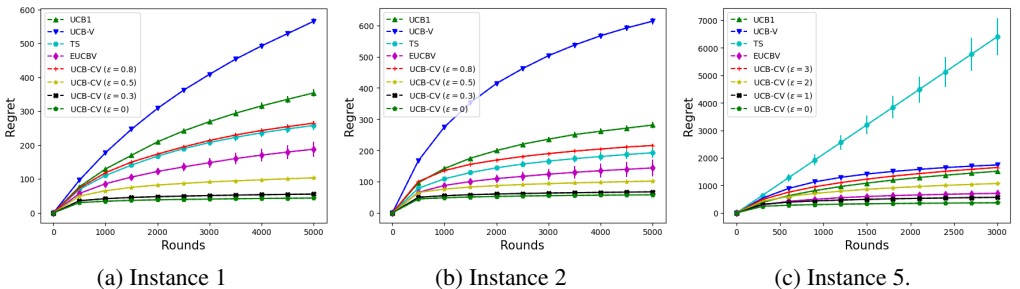

|     |     |     |
| :-: | :-: | :-: |
| (a) Instance 1 | (b) Instance 2 | (c) Instance 5. |

Figure 4: Regret comparison of UCB-CV with varying approximation error in the mean estimation of control variates and different multi-armed bandits algorithms.

Our experimental results demonstrate that as long as the error in the estimation of the mean of CVs is within a limit, there will be an advantage using control variates. The impact of such approximations in bandits needs to be analyzed extensively and demands independent work.

## 7   Conclusion

In this work, we studied stochastic multi-armed bandits problem with side information available in the form of Control Variate (CV). Leveraging the linear control variates' variance reduction properties, we developed a variant of the UCB algorithm named UCB-CV. When reward and CVs have a multivariate normal distribution, we showed that the regret of UCB-CV, which is of the order $\mathcal{O}(\sigma^2(1 - \rho^2)/\Delta \log T)$ where $\sigma^2$ is the maximum variance of arm rewards and $\rho$ is the minimum correlation between the arm rewards and CVs. As expected, our the bounds showed that when the correlation is strong, one can get significant improvement in regret performance. For the case where the reward and control variates follow a general distribution, we discussed Jackknifing and splitting resampling that will give estimators with smaller bias and sharper confidence bounds.

The variance reduction techniques based on CVs assume that the exact mean of the CVs are known. In practice, only some rough estimates of the mean of CVs rather than the exact values may be available. It would be interesting to study how does it affect the performance of the bandit algorithms. Another interesting direction to study how CVs are helpful if one has to rank the arms instead of just finding the top-ranked arm.

## Acknowledgements

Manjesh K. Hanawal is supported by INSPIRE faculty fellowship from DST and Early Career Research Award (ECR/2018/002953) from SERB, Govt. of India. This work was done when Arun Verma was at IIT Bombay.

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
