}^{\mathrm{J}} - \hat{\mu}_{s,i,q}^{c,\mathrm{J}})^2$. $\hat{\mu}_{s,i,q}^{c,\mathrm{J}}$ is shown to be unbiased (Nelson, 1989)[Thm. 7] when reward variance is bounded. Further, $\hat{\mu}_{s,i,q}^{c,\mathrm{J}} \pm t_{\alpha/2}(n-1)\hat{\nu}_{s,i,q}^{\mathrm{J}}$ is a asymptotically valid confidence interval (Avramidis et al., 1991) and holds approximately for finite number of samples.

## 5.2 Splitting

Splitting is a technique to split the correlated observations into two or more groups, compute an estimate of $\boldsymbol{\beta}_i^*$ from each group, then exchange the estimates among the groups. Nelson (1990) considers an extreme form of splitting, where he splits $s$ observations into $s$ groups. The $j^{\mathrm{th}}$ observation is given by

$$Y_{j,i,q}^{\mathrm{S}} = X_{j,i} + \hat{\boldsymbol{\beta}}_i^{*-j}(\boldsymbol{\omega} - \boldsymbol{W}_{j,i}), \quad j \in [s],$$

where $\hat{\boldsymbol{\beta}}_i^{*-j}$ is estimated without $j^{\mathrm{th}}$ observation and $\boldsymbol{W}_{j,i} = (W_{j,i,1}, \ldots, W_{j,i,q})$ is the vector of CVs with $q$ elements associated with reward $X_{j,i}$. The point estimator for splitting method is $\hat{\mu}_{s,i,q}^{c,\mathrm{S}} = \frac{1}{s}\sum_{j=1}^{s} Y_{j,i,q}^{\mathrm{S}}$ and its sample variance is $\hat{\nu}_{s,i,q}^{\mathrm{S}} = (s(s-1))^{-1}\sum_{j=1}^{s}(Y_{j,i,q}^{\mathrm{S}} - \hat{\mu}_{s,i,q}^{c,\mathrm{S}})^2$. Then $\hat{\mu}_{s,i,q}^{c,\mathrm{S}} \pm t_{\alpha/2}(n-1)\hat{\nu}_{s,i,q}^{\mathrm{S}}$ gives an approximate confidence interval (Nelson, 1990).

Below we define UCB index for these methods. Let $\hat{\nu}_{N_i(t),i,q}^{\mathrm{G}}$ be the sample variance of mean reward estimator $(\hat{\mu}_{s,i,q}^{c,\mathrm{G}})$ for arm $i$. We define the optimistic upper bound for mean reward for $G \in \{J, S\}$ as follows:

$$\mathrm{UCB}_{t,i,q}^{\mathrm{G}} = \hat{\mu}_{N_i(t),i,q}^{c,\mathrm{G}} + V_{t,N_i(t),q}^{\mathrm{G},\alpha}\sqrt{\hat{\nu}_{N_i(t),i,q}^{\mathrm{G}}}. \tag{6}$$

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

# Appendix

## A  Control Variates and Regression Theory

Consider the following regression problem with $n$ samples and $p$ features:

$$Y_i = \boldsymbol{X}_i^\top \boldsymbol{\theta} + \varepsilon_i, \quad i \in \{1, 2, \dots, n\}$$

where $Y_i \in \mathbb{R}$ is the $i^{\text{th}}$ target variable, $\boldsymbol{X}_i \in \mathbb{R}^p$ is the $i^{\text{th}}$ feature vector, $\boldsymbol{\theta} \in \mathbb{R}^p$ is the unknown regression parameters, and $\varepsilon_i$ is a normally distributed noise with mean $0$ and constant variance $\sigma^2$. The values of noise $\varepsilon_i$ form a IID sequence and are independent of $\boldsymbol{X}_i$. Let

$$\boldsymbol{Y} = \begin{pmatrix} Y_1 \\ \vdots \\ Y_n \end{pmatrix}, \qquad \boldsymbol{X}^\top = \begin{pmatrix} X_{11} & \dots & X_{1p} \\ \vdots & \dots & \vdots \\ X_{n1} & \dots & X_{np} \end{pmatrix}, \text{ and } \quad \boldsymbol{\varepsilon} = \begin{pmatrix} \varepsilon_1 \\ \vdots \\ \varepsilon_n \end{pmatrix}.$$

The least square estimator is given by $\hat{\boldsymbol{\theta}} = (\boldsymbol{X}\boldsymbol{X}^\top)^{-1}\boldsymbol{X}\boldsymbol{Y}$. Next we give the finite sample properties of $\hat{\boldsymbol{\theta}}$ that are useful to prove the regret upper bounds.

**Fact 1.** *The finite sample properties of the least square estimator of $\boldsymbol{\theta}$:*

1. $\mathbb{E}\left[\hat{\boldsymbol{\theta}}|\boldsymbol{X}\right] = \boldsymbol{\theta},$                          *(unbiased estimator)*

2. $Var(\hat{\boldsymbol{\theta}}|\boldsymbol{X}) = \sigma^2(\boldsymbol{X}\boldsymbol{X}^\top)^{-1},$ *and*       *(expression for the variance)*

3. $Var(\hat{\boldsymbol{\theta}}_i|\boldsymbol{X}) = \sigma^2(\boldsymbol{X}\boldsymbol{X}^\top)_{ii}^{-1},$          *(element-wise variance)*

*where* $(\boldsymbol{X}\boldsymbol{X}^\top)_{ii}^{-1}$ *is the* $ii-$*element of the matrix* $(\boldsymbol{X}\boldsymbol{X}^\top)^{-1}$.

The first two properties are derived form Proposition 1.1 of (Hayashi, 2000), whereas the third property is taken from (Van De Geer, 2005). Next we give the finite sample properties of estimator of variance $\sigma^2$.

**Fact 2.** *(Hayashi, 2000, Proposition 1.2) Let* $\hat{\sigma}^2 = \frac{1}{n-p}\sum_{i=1}^n (Y_i - \boldsymbol{X}_i^\top\hat{\boldsymbol{\theta}})^2$ *be estimator of* $\sigma^2$ *and* $n > p$ *(so that* $\hat{\sigma}^2$ *is well defined). Then* $\mathbb{E}\left[\hat{\sigma}^2|\boldsymbol{X}\right] = \sigma^2$ *which implies that* $\hat{\sigma}^2$ *is a unbiased estimator of* $\hat{\sigma}^2$.

Now recall the unknown parameter estimation problem with $q$ control variates and $t$ observations:

$$\bar{X}_s = X_s + \hat{\boldsymbol{\beta}}^{*\top}(\boldsymbol{\omega} - \boldsymbol{W}_s), \quad s \in \{1, 2, \dots, t\}.$$

where the subscript for arm index $i$ and number of control variates $q$ are dropped for simplicity. Under the assumption of multivariate normal distribution, we can write $\bar{X}_s$ as follows:

$$\bar{X}_s = \mu + \boldsymbol{\beta}^{*\top}(\boldsymbol{\omega} - \boldsymbol{W}_s) + \varepsilon_s,$$

where $\varepsilon_1, \dots, \varepsilon_t$ are IID normally distributed random variables with mean $0$ and variance $\sigma^2$. Let

$$\bar{\boldsymbol{X}} = \begin{pmatrix} \bar{X}_1 \\ \vdots \\ \bar{X}_t \end{pmatrix}, \qquad \boldsymbol{Y} = \begin{pmatrix} 1 & \omega_1 - W_{11} & \dots & \omega_q - W_{1q} \\ \vdots & & \vdots & \dots & \vdots \\ 1 & \omega_1 - W_{t1} & \dots & \omega_q - W_{tq} \end{pmatrix}, \quad \boldsymbol{\gamma} = \begin{pmatrix} \mu \\ \boldsymbol{\beta}^* \end{pmatrix}, \text{ and } \quad \boldsymbol{\varepsilon} = \begin{pmatrix} \varepsilon_1 \\ \vdots \\ \varepsilon_t \end{pmatrix}.$$

Now let the least square estimator of $\mu$ be $\hat{\mu}$ and $\beta^\star$ be $\hat{\boldsymbol{\beta}}^\star$. Then using Fact 1,

$$\text{Var}(\hat{\mu}) = \sigma^2(\boldsymbol{Y}^\top\boldsymbol{Y})_{11}^{-1}$$

where $(\boldsymbol{Y}^\top\boldsymbol{Y})_{11}^{-1}$ is the upper left most element of matrix $(\boldsymbol{Y}^\top\boldsymbol{Y})^{-1}$ (Schmeiser, 1982). Recall definitions of $\boldsymbol{S}_{W_iW_i}$, $\hat{\mu}_{t,i}^c$, and $\hat{\omega}_i$ (and ignore the arm index for now), Then after $t$ observations, the estimator of $\text{Var}(\hat{\mu})$ is given by

$$\hat{\nu}_t = \frac{Z_t \hat{\sigma}^2(t)}{t} \tag{7}$$

where $Z_t = \left(1 + \frac{(\hat{\boldsymbol{\omega}}_t - \boldsymbol{\omega})^\top \boldsymbol{S}_{WW}^{-1} (\hat{\boldsymbol{\omega}}_t - \boldsymbol{\omega})}{1 - 1/t}\right)$ and $\hat{\sigma}^2(t) = \frac{1}{t-q-1} \sum_{s=1}^{t} (\bar{X}_s - \hat{\mu}_t^c)^2$ (Nelson, 1990).
Using Fact 2, $\hat{\sigma}^2(t)$ is an unbiased estimator of $\sigma^2(t)$. Next, we give the fundamental results of control variates estimators.

**Fact 3.** *(Nelson, 1990, Theorem 1) Let $Z_s = (X_s, W_{s1}, \ldots, W_{sq})^\top$. Suppose that $\{Z_1, \ldots, Z_t\}$ are IID $(q+1)-$variate normally distributed vectors with mean vector $(\mu, \boldsymbol{\omega})^\top$ and variance $\Sigma_{ZZ}$. Then*

$$\mathbb{E}\left[\hat{\mu}_t^c\right] = \mu,$$

$$Var(\hat{\mu}_t^c) = \frac{t-2}{t-q-2}(1-\rho^2)Var(\hat{\mu}_t),$$

$$\mathbb{E}\left[\hat{\nu}_t\right] = Var(\hat{\mu}_t^c),$$

$$\mathbb{P}\left\{\hat{\mu}_t^c - V_{v,t,q}^{(\alpha)} \sqrt{\hat{\nu}_t} \le \mu\right\} \ge 1 - 1/v^\alpha, \text{ and}$$

$$\mathbb{P}\left\{\hat{\mu}_t^c + V_{v,t,q}^{(\alpha)} \sqrt{\hat{\nu}_t} \ge \mu\right\} \ge 1 - 1/v^\alpha,$$

*where $\rho^2 = \sigma_{X\boldsymbol{W}} \Sigma_{\boldsymbol{W}\boldsymbol{W}}^{-1} \sigma_{\boldsymbol{W}X}^\top / \sigma^2$ is the square of the multiple correlation coefficient, $\sigma^2 = Var(X)$, $\sigma_{X\boldsymbol{W}} = (Cov(X, W_1), \ldots, Cov(X, W_q))$, and $V_{v,t,q}^{(\alpha)}$ is the $100(1 - 1/v^\alpha)^{th}$ percentile value of the $t-$distribution with $t - q - 1$ degrees of freedom.*

## B  Missing proofs involving Estimator with One Control Variate

**Lemma 1.** *Let the reward and control variate of each arm have a multivariate normal distribution. After observing $s$ samples of reward and control variate from arm $i$, define $\hat{\nu}_{s,i} = \frac{Z_{s,i} \hat{\sigma}_{c,i}^2(s)}{s}$, where*

$$Z_{s,i} = \left(1 - \frac{(\sum_{r=1}^{s}(W_{r,i} - \omega_i))^2}{s \sum_{r=1}^{s}(W_{r,i} - \omega_i)^2}\right)^{-1} \quad \text{and} \quad \hat{\sigma}_{c,i}^2(s) = \frac{1}{s-2} \sum_{r=1}^{s}(\bar{X}_{r,i} - \hat{\mu}_{s,i}^c)^2,$$

*then $\hat{\nu}_{s,i}$ is an unbiased variance estimator of $\hat{\mu}_{s,i}^c$, i.e., $\mathbb{E}[\hat{\nu}_{s,i}] = Var(\hat{\mu}_{s,i}^c)$.*

*Proof.* For simplicity of notation, we are dropping arm index $i$ from the subscript of $W$ in this proof. Using Fact 1, we know the variance of estimator for arm $i$ with one control variate is

$$Var(\hat{\mu}_{s,i}^c) = \sigma_{c,i}^2 (\boldsymbol{Y}^\top \boldsymbol{Y})_{11}^{-1},$$

where $Var(\bar{X}_i) = \sigma_{c,i}^2$ and

$$\boldsymbol{Y} = \begin{pmatrix} 1 & \omega - W_1 \\ \vdots & \vdots \\ 1 & \omega - W_s \end{pmatrix}.$$

Now we compute the value of $(\boldsymbol{Y}^\top \boldsymbol{Y})_{11}^{-1}$. First we get the $\boldsymbol{Y}^\top \boldsymbol{Y}$ as follows:

$$\boldsymbol{Y}^\top \boldsymbol{Y} = \begin{pmatrix} 1 & \cdots & 1 \\ \omega - W_1 & \cdots & \omega - W_s \end{pmatrix} \begin{pmatrix} 1 & \omega - W_1 \\ \vdots & \vdots \\ 1 & \omega - W_s \end{pmatrix} = \begin{pmatrix} s & \sum_{r=1}^{s}(\omega - W_r) \\ \sum_{r=1}^{s}(\omega - W_r) & \sum_{r=1}^{s}(\omega - W_r)^2 \end{pmatrix}.$$

Now we the $(\boldsymbol{Y}^\top \boldsymbol{Y})^{-1}$ as follows:

$$(\boldsymbol{Y}^\top \boldsymbol{Y})^{-1} = \frac{1}{s \sum_{r=1}^{s}(\omega - W_r)^2 - \left(\sum_{r=1}^{s}(\omega - W_r)\right)^2} \begin{pmatrix} \sum_{r=1}^{s}(\omega - W_r)^2 & -\sum_{r=1}^{s}(\omega - W_r) \\ -\sum_{r=1}^{s}(\omega - W_r) & s \end{pmatrix}.$$

The value of $(\boldsymbol{Y}^\top \boldsymbol{Y})_{11}^{-1}$ given as

$$(\boldsymbol{Y}^\top \boldsymbol{Y})_{11}^{-1} = \frac{\sum_{r=1}^{s}(\omega - W_r)^2}{s \sum_{r=1}^{s}(\omega - W_r)^2 - (\sum_{r=1}^{s}(\omega - W_r))^2}$$

$$= \frac{1}{s} \frac{1}{1 - \frac{\left(\sum_{r=1}^{s}(\omega - W_r)\right)^2}{s \sum_{r=1}^{s}(\omega - W_r)^2}}$$

$$= \frac{1}{s} \left(1 - \frac{\left(\sum_{r=1}^{s}(\omega - W_r)\right)^2}{s \sum_{r=1}^{s}(\omega - W_r)^2}\right)^{-1}$$

$$= \frac{1}{s} \left(1 - \frac{\left(\sum_{r=1}^{s}(W_r - \omega)\right)^2}{s \sum_{r=1}^{s}(W_r - \omega)^2}\right)^{-1}$$

$$\implies (\boldsymbol{Y}^\top \boldsymbol{Y})_{11}^{-1} = \frac{Z_s}{s},$$

where $Z_s = \left(1 - \frac{\left(\sum_{r=1}^{s}(W_r - \omega)\right)^2}{s \sum_{r=1}^{s}(W_r - \omega)^2}\right)^{-1}$. After $s$ number of observations of rewards and associated control variates from arm $i$, the estimator of $\mathrm{Var}(\hat{\mu})$ is given by

$$\hat{\nu}_{s,i} = \frac{Z_s \hat{\sigma}_{c,i}^2(s)}{s}$$

where $\hat{\sigma}_{c,i}^2(s) = \frac{1}{s-2} \sum_{r=1}^{s}(\bar{X}_r - \hat{\mu}_{s,i}^c)^2$ is the unbiased estimator of $\sigma_{c,i}^2$ (by Fact 2). Further, $\mathbb{E}[\hat{\nu}_{s,i}] = \mathrm{Var}(\hat{\mu}_{s,i}^c)$ (by Fact 3) which implies that $\hat{\nu}_{s,i}$ is an unbiased estimator of $\mathrm{Var}(\hat{\mu}_{s,i}^c)$. $\qquad\square$

Theorem 1 of Nelson (1990) shows that the empirical mean with control variate is an unbiased estimator for a given number of samples (as stated in Lemma 3). Our next result adapts the concentration bound given in Theorem 1 of Nelson (1990) for a given number of samples.

**Lemma 2.** *Let the conditions in Lemma 1 hold and $s$ be the number of reward and associated control variate samples from arm $i$ in round $t$. Then*

$$\mathbb{P}\left\{|\hat{\mu}_{s,i}^c - \mu_i| \geq V_{t,s}^{(\alpha)} \sqrt{\hat{\nu}_{s,i}}\right\} \leq 2/t^\alpha,$$

*where $V_{t,s}^{(\alpha)}$ denote $100(1 - 1/t^\alpha)^{th}$ percentile value of the $t-$distribution with $s - 2$ degrees of freedom and $\hat{\nu}_{s,i}$ is an unbiased estimator for variance of $\hat{\mu}_{s,i}^c$.*

*Proof.* The proof follows from Fact 3 with $q = 1$ control variates and replacing other parameters with arms specific parameters for $s$ observations of arm rewards and associated control variate, i.e., $\hat{\mu}_s^c$ by $\hat{\mu}_{s,i}^c$, $\mu$ by $\mu_i$, and $\hat{\nu}_s$ by $\hat{\nu}_{s,i}$. Note that we use $t-$distribution for confidence intervals, hence the value of $\mathbb{P}\left\{|\hat{\mu}_{s,i}^c - \mu_i| \geq V_{t,s,1}^{(\alpha)} \sqrt{\hat{\nu}_{s,i}}\right\}$ depends only on the value of $V_{t,s,1}^{(\alpha)}$. Now using following simple algebraic manipulations, we have

$$\mathbb{P}\left\{|\hat{\mu}_{s,i}^c - \mu_i| \geq V_{t,s,1}^{(\alpha)} \sqrt{\hat{\nu}_{s,i}}\right\} = 1 - \mathbb{P}\left\{|\hat{\mu}_{s,i}^c - \mu_i| \leq V_{t,s,1}^{(\alpha)} \sqrt{\hat{\nu}_{s,i}}\right\}$$

$$\leq 1 - \left(1 - \frac{2}{t^\alpha}\right) \qquad \text{(using Fact 3)}$$

$$\implies \mathbb{P}\left\{|\hat{\mu}_{s,i}^c - \mu_i| \geq V_{t,s,1}^{(\alpha)} \sqrt{\hat{\nu}_{s,i}}\right\} \leq 2/t^\alpha. \qquad\square$$

## C   Estimator with Multiple Control Variates

In this section, we extend our setup to case where the multiple control variates can be associated with an arm's rewards. Let $q$ be the number of control variates. Then the sample for arm $i$ with associated control variates is given by

$$\bar{X}_{t,i,q} = X_{t,i} + \sum_{j=1}^{q} \beta_{i,j}^*(\omega_i - W_{t,i,j}),$$

where $W_{t,i,j}$ is the $j^{\text{th}}$ control variate of arm $i$ that is observed in round $t$, $\omega_{i,j} = \mathbb{E}[W_{t,i,j}]$, and $\beta_{i,j}^* = \mathrm{Cov}(X_i, W_{i,j})/\mathrm{Var}(W_{i,j})$. The unbiased mean reward estimator for arm $i$ with $s$ samples of

rewards and associated $q$ number of control variates with known values of $\beta_{i,j}^*$ is given by

$$\hat{\mu}_{s,i,q}^c = \frac{1}{s}\sum_{r=1}^{s}\bar{X}_{r,i}.$$

Let $\hat{\mu}_{s,i} = \frac{1}{s}\sum_{r=1}^{s}X_{r,i}, \hat{\boldsymbol{\beta}}_i^* = \left(\hat{\beta}_{i,1},\ldots,\hat{\beta}_{i,q}\right)^{\top}$ is the estimate of $\boldsymbol{\beta}_i^*$, $\boldsymbol{\omega}_i = (\omega_{i,1},\ldots,\omega_{i,q})^{\top}$, and $\hat{\boldsymbol{\omega}}_{s,i} = (\hat{\omega}_{s,i,1},\ldots,\hat{\omega}_{s,i,q})^{\top}$, where $\hat{\omega}_{s,i,j} = \frac{1}{s}\sum_{r=1}^{s}W_{r,i,j}$. Then $\hat{\mu}_{s,i,q}^c$ can be written as:

$$\hat{\mu}_{s,i,q}^c = \hat{\mu}_{s,i} + \hat{\boldsymbol{\beta}}_i^{*\top}(\boldsymbol{\omega}_i - \hat{\boldsymbol{\omega}}_{s,i}), \tag{8}$$

Let $s$ be the number of rewards and associated control variates samples for arm $i$, $\boldsymbol{W}_i$ be the $s \times q$ matrix whose $r^{\text{th}}$ row is $(W_{r,i,1}, W_{r,i,2}, \ldots, W_{r,i,q})$, $\boldsymbol{S}_{W_iW_i} = (s-1)^{-1}(\boldsymbol{W}_i^{\top}\boldsymbol{W}_i - s\hat{\boldsymbol{\omega}}_{s,i}\hat{\boldsymbol{\omega}}_{s,i}^{\top})$, and $\boldsymbol{S}_{X_iW_i} = (s-1)^{-1}(\boldsymbol{W}_i^{\top}\boldsymbol{X}_i - s\hat{\boldsymbol{\omega}}_i\,\hat{\mu}_{s,i})$ where $\boldsymbol{X}_i = (X_{1,i},\ldots,X_{s,i})^{\top}$. Then by extending the arguments used in Eq. (4) to get estimated coefficient for a scalar $\hat{\beta}_i^*$ to a vector $\hat{\boldsymbol{\beta}}_i^*$, the estimated coefficient vector is given by

$$\hat{\boldsymbol{\beta}}_i^* = \boldsymbol{S}_{W_iW_i}^{-1}\boldsymbol{S}_{X_iW_i}. \tag{9}$$

Our next results are generalization of Lemma 1 and Lemma 2 to MAB-CV problems with $q$ control variates.

**Lemma 4.** *Let $s$ be the number of rewards and associated control variate samples. Then, the sample variance of mean reward estimator for arm $i$ is given by*

$$\hat{\nu}_{s,i,q} = \frac{Z_{s,q}\hat{\sigma}_{c,i,q}^2(s)}{s}, \quad \text{where } Z_{s,q} = \left(1 + \frac{(\hat{\boldsymbol{\omega}}_{s,i} - \boldsymbol{\omega}_i)^{\top}\boldsymbol{S}_{W_iW_i}^{-1}(\hat{\boldsymbol{\omega}}_{s,i} - \boldsymbol{\omega}_i)}{1 - 1/s}\right)$$

*and $\hat{\sigma}_{c,i}^2(s) = \frac{1}{s-q-1}\sum_{r=1}^{s}(\bar{X}_r - \hat{\mu}_{s,i}^c)^2$. Further, $\hat{\nu}_{s,i,q}$ is unbiased estimator.*

*Proof.* Using Fact 1, we know the variance of estimator for arm $i$ with one control variate is

$$\text{Var}(\hat{\mu}_{s,i}^c) = \sigma_{c,i}^2(\boldsymbol{Y}^{\top}\boldsymbol{Y})_{11}^{-1}.$$

After having $s$ observations of rewards and associated $q$ control variates from arm $i$, we use Eq. (7) to get an unbiased estimator for $\text{Var}(\hat{\mu}_{s,i}^c)$ as follows:

$$\hat{\nu}_{s,i,q} = \frac{Z_{s,q}\hat{\sigma}_{c,i,q}^2(s)}{s} \tag{10}$$

where $Z_{s,q} = \left(1 + \frac{(\hat{\boldsymbol{\omega}}_{s,i} - \boldsymbol{\omega}_i)^{\top}\boldsymbol{S}_{W_iW_i}^{-1}(\hat{\boldsymbol{\omega}}_{s,i} - \boldsymbol{\omega}_i)}{1 - 1/s}\right)$ and $\hat{\sigma}_{c,i,q}^2(s) = \frac{1}{s-q-1}\sum_{r=1}^{s}(\bar{X}_r - \hat{\mu}_{s,i}^c)^2$ (Nelson, 1990). Also note that $\hat{\sigma}_{c,i,q}^2(t)$ is an unbiased estimator of $\text{Var}(\bar{X}_i)$ (by Fact 2). $\qquad\square$

**Lemma 5.** *Let $s$ be the number of rewards and associated control variates samples from arm $i$ in round $t$. Then*

$$\mathbb{P}\left\{|\hat{\mu}_{s,i,q}^c - \mu_i| \geq V_{t,s,q}^{(\alpha)}\sqrt{\hat{\nu}_{s,i,q}}\right\} \leq 2/t^{\alpha}.$$

*where $V_{t,s,q}^{(\alpha)}$ is the $100(1 - 1/t^{\alpha})^{th}$ percentile value of the $t-$distribution with $s - q - 1$ degrees of freedom and $\hat{\nu}_{s,i,q}$ is an unbiased estimator for variance of $\hat{\mu}_{s,i,q}^c$.*

*Proof.* The proof follows from Fact 3 with $q$ control variates and replacing other parameters with arms specific parameters for $s$ observations of arm rewards and associated control variate, i.e., $\hat{\mu}_s^c$ by $\hat{\mu}_{s,i}^c$, $\mu$ by $\mu_i$, and $\hat{\nu}_s$ by $\hat{\nu}_{s,i}$. As we use $t-$distribution for confidence intervals, the value of $\mathbb{P}\left\{|\hat{\mu}_{t,i,q}^c - \mu_i| \geq V_{t,s,q}^{(\alpha)}\sqrt{\hat{\nu}_{s,q}}\right\}$ depends only on the value of $V_{t,s,q}^{(\alpha)}$. Now using following simple algebraic manipulations, we have

$$\mathbb{P}\left\{|\hat{\mu}_{s,i,q}^c - \mu_i| \geq V_{t,s,q}^{(\alpha)}\sqrt{\hat{\nu}_{s,i,q}}\right\} = 1 - \mathbb{P}\left\{|\hat{\mu}_{s,i,q}^c - \mu_i| \leq V_{t,s,q}^{(\alpha)}\sqrt{\hat{\nu}_{s,i,q}}\right\}$$

$$\leq 1 - \left(1 - \frac{2}{t^\alpha}\right) \qquad \text{(using Fact 3)}$$

$$\implies \mathbb{P}\left\{|\hat{\mu}_{s,i,q}^c - \mu_i| \geq V_{t,s,q}^{(\alpha)}\sqrt{\hat{\nu}_{s,i,q}}\right\} \leq 2/t^\alpha. \qquad \qquad \square$$

Next, we define optimistic upper bound for estimate of mean reward with $q$ control variates as follows:

$$\text{UCB}_{t,i,q} = \hat{\mu}_{N_i(t),i,q}^c + V_{t,N_i(t),q}^{(\alpha)}\sqrt{\hat{\nu}_{N_i(t),i,q}}. \qquad (11)$$

For multiple control variate case, we can use UCB-CV with $Q = q + 2$ and replacing $\text{UCB}_{t,i}$ by $\text{UCB}_{t,i,q}$ as defined in Eq. (11).

## D  Regret Analysis of UCB-CV

Similar to Bubeck et al. (2012), our following result defines three events and shows that the sub-optimal arm is only selected if one of these events are true.

**Lemma 6.** *Let $N_i(t)$ be the number of times sub-optimal arm selected until $t$ rounds. A sub-optimal arm $i$ is selected by UCB-CV in round $t$ if at least one of the three following events must be true:*

1. $\hat{\mu}_{N_i^\star(t),i^\star,q}^c + V_{t,N_i^\star(t),q}^{(\alpha)}\sqrt{\hat{\nu}_{N_i^\star(t),i^\star,q}} \leq \mu_{i^\star}$,

2. $\hat{\mu}_{N_i(t),i,q}^c - V_{t,N_i(t),q}^{(\alpha)}\sqrt{\hat{\nu}_{N_i(t),i,q}} > \mu_i$, *and*

3. $N_i(t) < \dfrac{4(V_{T,N_i(t),q}^{(\alpha)})^2 Z_{N_i(t),q}\hat{\sigma}_{c,i,q}^2(N_i(t))}{\Delta_i^2}$.

*Proof.* Let assume that all three events are all false, then we have:

$$\text{UCB}_{t,i^\star} = \hat{\mu}_{N_i^\star(t),i^\star,q}^c + V_{t,N_i^\star(t),q}^{(\alpha)}\sqrt{\hat{\nu}_{N_i^\star(t),i^\star,q}}$$

$$> \mu_{i^\star} \qquad \text{(if event 1 is false)}$$

$$= \mu_i + \Delta_i$$

$$\geq \mu_i + 2V_{T,N_i(t),q}^{(\alpha)}\sqrt{\frac{Z_{N_i(t),q}\hat{\sigma}_{c,i,q}^2(N_i(t))}{N_i(t)}}. \qquad \text{(if event 3 is false)}$$

From Lemma 4, we have $\hat{\nu}_{N_i(t),i,q} = Z_{N_i(t),q}\hat{\sigma}_{c,i,q}^2(N_i(t))/N_i(t)$

$$\text{UCB}_{t,i^\star} > \mu_i + 2V_{T,N_i(t),q}^{(\alpha)}\sqrt{\hat{\nu}_{N_i(t),i,q}}$$

$$\geq \mu_i + 2V_{t,N_i(t),q}^{(\alpha)}\sqrt{\hat{\nu}_{N_i(t),i,q}} \qquad \left(V_{t,N_i(t),q}^{(\alpha)} \text{ is increasing function of } t \text{ and } t \leq T\right)$$

$$\geq \hat{\mu}_{N_i(t),i,q}^c - V_{t,N_i(t),q}^{(\alpha)}\sqrt{\hat{\nu}_{N_i(t),i,q}} + 2V_{t,N_i(t),q}^{(\alpha)}\sqrt{\hat{\nu}_{N_i(t),i,q}} \qquad \text{(if event 2 is false)}$$

$$= \hat{\mu}_{N_i(t),i,q}^c + V_{t,N_i(t),q}^{(\alpha)}\sqrt{\hat{\nu}_{N_i(t),i,q}}$$

$$= \text{UCB}_{t,i}$$

$$\implies \text{UCB}_{t,i^\star} > \text{UCB}_{t,i}.$$

If all three events are false, then the optimal arm's UCB value is larger than the UCB value of the sub-optimal arm $i$. Therefore, at least one of three events need to be true if the sub-optimal arm $i$ is selected in round $t$. It concludes our proof. $\qquad \square$

When the number of samples is random, randomness has to be taken into account. The following result gives the concentration bound on empirical mean with the random number of observations.

**Lemma 7.** *Let $t \in \mathbb{N}$, $\mu$ be the true mean, $\hat{\mu}_s$ be the empirical mean with $s(\leq t)$ observations, and $f(t,s) = V_{t,s,q}^{(2)}\sqrt{\hat{\nu}_{s,i,q}}$. If $\mathbb{P}\{|\hat{\mu}_s - \mu| \leq f(t,s)\} \leq 1/t^2$ and $N_t$ is the random number such that $N_t \in [t]$, then*

$$\mathbb{P}\{|\hat{\mu}_{N_t} - \mu| \leq f(t, N_t)\} \leq \frac{1}{t^2}.$$

*Proof.* Using the conditioning argument, we have

$$\mathbb{P}\left\{|\hat{\mu}_{N_t} - \mu| \le f(t, N_t)\right\} = \sum_{s=1}^{t} \mathbb{P}\left\{|\hat{\mu}_s - \mu| \le f(t, s)|N_t = s\right\} \mathbb{P}\left\{N_t = s\right\}$$

$$\le \sum_{s=1}^{t} \frac{1}{t^2} \mathbb{P}\left\{N_t = s\right\} \qquad \left(\text{as } \mathbb{P}\left\{|\hat{\mu}_s - \mu| \le f(t, s)\right\} \le 1/t^2\right)$$

$$= \frac{1}{t^2} \sum_{s=1}^{t} \mathbb{P}\left\{N_t = s\right\}.$$

As $N_t \in [t]$, $\sum_{s=1}^{t} \mathbb{P}\left\{N_t = s\right\} = 1$, we have $\mathbb{P}\left\{|\hat{\mu}_{N_t} - \mu| \le f(t, N_t)\right\} \le \frac{1}{t^2}$. $\qquad \square$

**Lemma 3.** *Let reward and control variates have a multivariate normal distribution and $q$ be the number of control variates. Then after having $s$ observations of rewards and associated control variates from arm $i$,*

$$\mathbb{E}\left[\hat{\mu}_{s,i,q}^c\right] = \mu_i, \text{ and}$$

$$Var(\hat{\mu}_{s,i,q}^c) = \frac{s-2}{s-q-2}(1 - \rho_i^2)Var(\hat{\mu}_{s,i}),$$

*where $\rho_i^2 = \sigma_{X_i \boldsymbol{W}_i} \Sigma_{\boldsymbol{W}_i \boldsymbol{W}_i}^{-1} \sigma_{X_i \boldsymbol{W}_i}^{\top}/\sigma_i^2$ is the square of the multiple correlation coefficient, $\sigma_i^2 = Var(X_i)$, and $\sigma_{X_i \boldsymbol{W}_i} = (Cov(X_i, W_{i,1}), \ldots, Cov(X_i, W_{i,q}))$.*

*Proof.* From Fact 3, we know that the estimator $(\hat{\mu}_{s,i,q}^c)$ and its variance estimator $(\hat{\nu}_{s,i,q})$ are unbiased estimator. Further, using Fact 3, we can get the expression mentioned in Lemma 3 for the variance of reward estimator $(\hat{\nu}_{s,i,q})$. $\qquad \square$

Using Lemma 3, now we can upper bound the number of times a sub-optimal arm $i$ is selected by UCB-CV in the following result.

**Lemma 8.** *Let $\alpha = 2$, $q$ be the number of control variates, and $N_i(T)$ be the number of times sub-optimal arm $i$ selected in $T$ rounds. If $C_{T,i,q} = \mathbb{E}\left[\frac{N_i(T)-2}{N_i(T)-q-2}\left(\frac{V_{T,N_i(T),q}^{(2)}}{V_{T,T,q}^{(2)}}\right)^2\right]$ then the expected number of times a sub-optimal arm $i \in [K]$ selected by UCB-CV in $T$ rounds is upper bounded by*

$$\mathbb{E}\left[N_i(T)\right] \le \frac{4(V_{T,T,q}^{(2)})^2 C_{T,i,q}(1 - \rho_i)\sigma_i^2}{\Delta_i^2} + \frac{\pi^2}{3} + 1.$$

*Proof.* Let $u = \left\lceil \frac{4(V_{T,N_i(T),q}^{(2)})^2 Z_{N_i(T),q}\hat{\sigma}_{c,i,q}^2(N_i(T))}{\Delta_i^2}\right\rceil$ then the Event 3 is false for $N_i(T) = u$. Further the value of $u$ is random as its depends on $V_{T,N_i(T),q}^{(2)}$, $Z_{N_i(T),q}$, and $\hat{\sigma}_{c,i,q}^2(N_i(T))$. Now using Lemma 6, we have

$$\mathbb{E}\left[N_i(T)\right] = \mathbb{E}\left[\sum_{t=1}^{T} \mathbb{1}_{\{I_t=i\}}\right]$$

$$\le \mathbb{E}\left[\sum_{t=1}^{T} \mathbb{1}_{\{\text{Event 3 is true}\}}\right] + \mathbb{E}\left[\sum_{t=1}^{T} \mathbb{1}_{\{I_t=i \text{ and Event 3 is false}\}}\right]$$

$$\le \mathbb{E}\left[u\right] + \mathbb{E}\left[\sum_{t=1}^{T} \mathbb{1}_{\{\text{Either Event 1 or Event 2 is true}\}}\right]$$

$$\implies \mathbb{E}\left[N_i(T)\right] \le \mathbb{E}\left[u\right] + \mathbb{E}\left[\sum_{t=1}^{T} \mathbb{1}_{\{\text{Event 1 is true}\}}\right] + \mathbb{E}\left[\sum_{t=1}^{T} \mathbb{1}_{\{\text{Event 2 is true}\}}\right]. \qquad (12)$$

We will first bound $\mathbb{E}\left[\sum_{t=1}^{T}\mathbb{1}_{\{\text{Event 1 is true}\}}\right]$ as follows

$$\mathbb{E}\left[\sum_{t=1}^{T}\mathbb{1}_{\{\text{Event 1 is true}\}}\right] = \sum_{t=1}^{T}\mathbb{P}\left\{\text{Event 1 is true}\right\}$$

$$= \sum_{t=1}^{T}\mathbb{P}\left\{\hat{\mu}_{N_i^{\star}(t),i^{\star},q}^{c} + V_{t,N_i(t),q}^{(2)}\sqrt{\hat{\nu}_{N_i^{\star}(t),i^{\star},q}} \leq \mu_{i^{\star}}\right\}$$

$$= \sum_{t=1}^{T}\mathbb{P}\left\{\hat{\mu}_{N_i^{\star}(t),i^{\star},q}^{c} - \mu_{i^{\star}} \leq -V_{t,N_i(t),q}^{(2)}\sqrt{\hat{\nu}_{N_i^{\star}(t),i^{\star},q}}\right\}$$

$$= \sum_{t=1}^{T}\frac{1}{t^2} \qquad\qquad\text{(using Lemma 5 and Lemma 7)}$$

$$\leq \sum_{t=1}^{\infty}\frac{1}{t^2}$$

$$\implies \mathbb{E}\left[\sum_{t=1}^{T}\mathbb{1}_{\{\text{Event 1 is true}\}}\right] \leq \frac{\pi^2}{6}. \tag{13}$$

Now we will bound $\mathbb{E}\left[\sum_{t=1}^{T}\mathbb{1}_{\{\text{Event 2 is true}\}}\right]$ as follows

$$\mathbb{E}\left[\sum_{t=1}^{T}\mathbb{1}_{\{\text{Event 2 is true}\}}\right] = \sum_{t=1}^{T}\mathbb{P}\left\{\text{Event 2 is true}\right\}$$

$$= \sum_{t=1}^{T}\mathbb{P}\left\{\hat{\mu}_{N_i(t),i,q}^{c} - V_{t,N_i(t),q}^{(2)}\sqrt{\hat{\nu}_{N_i(t),i,q}} > \mu_i\right\}$$

$$= \sum_{t=1}^{T}\mathbb{P}\left\{\hat{\mu}_{N_i(t),i,q}^{c} - \mu_i > V_{t,N_i(t),q}^{(2)}\sqrt{\hat{\nu}_{N_i(t),i,q}}\right\}$$

$$= \sum_{t=1}^{T}\frac{1}{t^2} \qquad\qquad\text{(using Lemma 5 and Lemma 7)}$$

$$\leq \sum_{t=1}^{\infty}\frac{1}{t^2}$$

$$\implies \mathbb{E}\left[\sum_{t=1}^{T}\mathbb{1}_{\{\text{Event 1 is true}\}}\right] \leq \frac{\pi^2}{6}. \tag{14}$$

Last, we will bound $\mathbb{E}\left[u\right]$ as follows

$$\mathbb{E}\left[u\right] = \mathbb{E}\left[\left\lceil\frac{4(V_{T,N_i(T),q}^{(2)})^2 Z_{N_i(T),q}\hat{\sigma}_{c,i,q}^2(N_i(T))}{\Delta_i^2}\right\rceil\right]$$

$$\leq \mathbb{E}\left[\frac{4(V_{T,N_i(T),q}^{(2)})^2 Z_{N_i(T),q}\hat{\sigma}_{c,i,q}^2(N_i(T))}{\Delta_i^2}\right] + 1$$

$$= \frac{4}{\Delta_i^2}\mathbb{E}\left[(V_{T,N_i(T),q}^{(2)})^2 Z_{N_i(T),q}\hat{\sigma}_{c,i,q}^2(N_i(T))\right] + 1.$$

Divide and multiply LHS by $(V_{T,T,q}^{(2)})^2$ and $N_i(T)$, we have

$$\mathbb{E}\left[u\right] \leq \frac{4(V_{T,T,q}^{(2)})^2}{\Delta_i^2}\mathbb{E}\left[\left(\frac{V_{T,N_i(T),q}^{(2)}}{V_{T,T,q}^{(2)}}\right)^2\frac{Z_{N_i(T),q}\hat{\sigma}_{c,i,q}^2(N_i(T))}{N_i(T)}N_i(T)\right] + 1.$$

Using $\hat{\nu}_{N_i(T),i,q} = \frac{Z_{N_i(T),q}\hat{\sigma}^2_{c,i,q}(N_i(T))}{N_i(T)}$ and the Law of Iterated Expectation, we get

$$\mathbb{E}\left[u\right] \leq \frac{4(V^{(2)}_{T,T,q})^2}{\Delta_i^2}\mathbb{E}\left[\left(\frac{V^{(2)}_{T,N_i(T),q}}{V^{(2)}_{T,T,q}}\right)^2 \hat{\nu}_{N_i(T),i,q}N_i(T)\right] + 1$$

$$= \frac{4(V^{(2)}_{T,T,q})^2}{\Delta_i^2}\mathbb{E}\left[\mathbb{E}\left[\left(\frac{V^{(2)}_{T,N_i(T),q}}{V^{(2)}_{T,T,q}}\right)^2 \hat{\nu}_{N_i(T),i,q}N_i(T)|N_i(T)\right]\right] + 1$$

$$\implies \mathbb{E}\left[u\right] = \frac{4(V^{(2)}_{T,T,q})^2}{\Delta_i^2}\mathbb{E}\left[\left(\frac{V^{(2)}_{T,N_i(T),q}}{V^{(2)}_{T,T,q}}\right)^2 N_i(T)\mathbb{E}\left[\hat{\nu}_{N_i(T),i,q}|N_i(T)\right]\right] + 1.$$

We know that $\hat{\nu}_{N_i(T),i,q}$ given $N_i(T)$ is an unbiased estimator (Lemma 4) of variance of reward estimator that used $N_i(T)$ observations. Further, the reward observations are IID hence $\text{Var}(\hat{\mu}_{s,i}) = \sigma_i^2/s$ where $\sigma_i^2 = \text{Var}(X_{s,i})$. With this, using Lemma 3 for getting $\mathbb{E}\left[\hat{\nu}_{N_i(T),i,q}|N_i(T)\right]$, we have

$$\mathbb{E}\left[u\right] = \frac{4(V^{(2)}_{T,T,q})^2}{\Delta_i^2}\mathbb{E}\left[\left(\frac{V^{(2)}_{T,N_i(T),q}}{V^{(2)}_{T,T,q}}\right)^2 N_i(T)\frac{N_i(T)-2}{N_i(T)-q-2}(1-\rho_i^2)\frac{\sigma^2}{N_i(T)}\right] + 1$$

$$= \frac{4(V^{(2)}_{T,T,q})^2(1-\rho_i^2)\sigma_i^2}{\Delta_i^2}\mathbb{E}\left[\frac{N_i(T)-2}{N_i(T)-q-2}\left(\frac{V^{(2)}_{T,N_i(T),q}}{V^{(2)}_{T,T,q}}\right)^2\right] + 1.$$

Since $C_{T,i,q} = \mathbb{E}\left[\frac{N_i(T)-2}{N_i(T)-q-2}\left(\frac{V^{(2)}_{T,N_i(T),q}}{V^{(2)}_{T,T,q}}\right)^2\right]$, we have

$$\implies \mathbb{E}\left[u\right] \leq \frac{4(V^{(2)}_{T,T,q})^2 C_{T,i,q}(1-\rho_i^2)\sigma_i^2}{\Delta_i^2} + 1. \tag{15}$$

Using Eq. (13), Eq. (14), and Eq. (15), in Eq. (12), we get

$$\mathbb{E}\left[N_i(T)\right] \leq \frac{4(V^{(2)}_{T,T,q})^2 C_{T,i,q}(1-\rho_i^2)\sigma_i^2}{\Delta_i^2} + \frac{\pi^2}{3} + 1. \qquad \square$$

Now we are ready to upper bound the regret of UCB-CV.

**Theorem 1.** *Let the conditions in Lemma 3 hold, $\alpha = 2$, $\Delta_i = \mu_{i^\star} - \mu_i$ be the sub-optimality gap for arm $i \in [K]$, and $N_i(T)$ be the number of times sub-optimal arm $i$ selected in $T$ rounds. Let $C_{T,i,q} = \mathbb{E}\left[\frac{N_i(T)-2}{N_i(T)-q-2}\left(\frac{V^{(2)}_{T,N_i(T),q}}{V^{(2)}_{T,T,q}}\right)^2\right]$ for all $i$. Then the regret of UCB-CV in $T$ rounds is upper bounded by*

$$\Re_T \leq \sum_{i \neq i^\star}\left(\frac{4(V^{(2)}_{T,T,q})^2 C_{T,i,q}(1-\rho_i^2)\sigma_i^2}{\Delta_i} + \frac{\Delta_i\pi^2}{3} + \Delta_i\right).$$

*Proof.* By definition, we have

$$\Re_T = T\mu_{i^\star} - \mathbb{E}\left[\sum_{t=1}^{T} X_{t,I_t}\right] = \sum_{i=1}^{K}\Delta_i\mathbb{E}\left[N_i(T)\right].$$

Using Lemma 8 to replace $\mathbb{E}\left[N_i(T)\right]$, we get

$$\Re_T \leq \sum_{i \in [K]\backslash\{i^\star\}}\left(\frac{4(V^{(2)}_{T,T,q})^2 C_{T,i,q}(1-\rho_i^2)\sigma_i^2}{\Delta_i} + \frac{\Delta_i\pi^2}{3} + \Delta_i\right). \qquad \square$$

# E   Batching

The batching is well known method used for calculating confidence intervals for means of a sequence of correlated observations (Conway, 1963; Schmeiser, 1982). Batching transforms correlated observations into smaller uncorrelated and (almost) normally distributed observations called batch means. Let $\bar{X}_{1,i}, \bar{X}_{2,i}, \ldots, \bar{X}_{s,i}$ be the correlated observations from arm $i \in [K]$. These can be transformed into $b$ batch means, where each batch mean uses $m$ correlated observations. The value of $b = \lfloor s/m \rfloor$. The $j^{\text{th}}$ mean batch is given by

$$Y_{j,i,q}^{\text{B}} = \frac{1}{m} \sum_{i=(j-1)m+1}^{jm} \bar{X}_{s,i,q}, \quad j \in \{1, 2, \ldots, b\}.$$

Let the mean reward estimator be $\hat{\mu}_{s,i,q}^{c,\text{B}} = \frac{1}{\lfloor s/m \rfloor} \sum_{j=1}^{\lfloor s/m \rfloor} Y_{j,i,q}^{\text{B}}$ and $\hat{\nu}_{N_i(t),i,q}^{\text{B}}$ be the sample variance of mean reward estimator for arm $i$. Under assumption that batch means are normally distributed, we can define the optimistic upper bound for mean reward estimate as follows:

$$\text{UCB}_{t,i,q}^{\text{B}} = \hat{\mu}_{M_i(t),i,q}^{c,\text{B}} + V_{t,M_i(t),q}^{\text{B},\alpha} \sqrt{\hat{\nu}_{N_i(t),i,q}^{\text{B}}},$$

where $M_i(t)$ is the number of batch means until round $t$, $V_{t,s,q}^{\text{B},\alpha}$ is the $100(1 - 1/t^\alpha)^{\text{th}}$ percentile value of the $t-$distribution with $\lfloor s/m \rfloor - q - 1$ degrees of freedom, and $\hat{\nu}_{s,i,q}^{\text{G}}$ is computed using Lemma 4 with $\lfloor s/m \rfloor$ observations. We can use UCB-CV with $Q = mq - q + 1$ and replacing $\text{UCB}_{t,i}$ by $\text{UCB}_{t,i,q}^{\text{B}}$ for solving the MAB-CV problem instance using batching method.

The cost for batching is the loss of degrees of freedom. Nelson (1989) quantifies this cost in terms of the variance estimator when $\{\bar{X}_i\}$ is supposed to be having a multivariate normal distribution. Unfortunately, this assumption only holds for a minimum batch size $(m^\star)$ that makes batch means normally distributed uncorrelated observations. The value of $m^\star$ depends on the observations and there exists no closed-form expression for it.

We refer the readers to Nelson (1989) to know how to select the batch size in batching method. More detailed discussion about batching, jackknifing, and splitting can be found in Nelson (1990).