# OpenReview forum: "Stochastic Multi-Armed Bandits with Control Variates"
_NeurIPS.cc/2021/Conference — NeurIPS 2021 Poster_

### Official Review · Reviewer_2Zfv · 2021-06-28

**Rating:** 6
**Confidence:** 3

**Summary:**

This paper considers the (stochastic) multi-armed bandit problem, and proposes to use control variates to improve estimates of the reward for each arm. The case of Gaussian reward and control variates is studied in detail from a theoretical viewpoint, and the more general case is studied empirically.

**Limitations And Societal Impact:**

There was no discussion of societal impact. I have also highlighted above a number of limitations which should be discussed in more details.

**Main Review:**

I should clarify before starting my review that my knowledge of reinforcement learning is fairly limited, but that I have significant experience with the use of control variates for Monte Carlo methods more generally. My comments should therefore be interpreted with this in mind.


Main comments
----------------

- I found the paper to be well written and argued. This made it easier for me to read given I am not an expert in reinforcement learning.
- The literature review seems fairly thorough to me. A large number of relevant papers are discussed, and comparisons are made with the present paper. I have nevertheless suggested a number of other papers from the statistics literature which could be interesting to discuss, mostly as I believe they could potentially bring further gains. See the "literature" section below.
- Section 4: It is very much unclear to me whether assuming that the reward and control variate of each arm have a multivariate normal distribution is something reasonable in practice. From the literature review (Section 1.1.), it seems like a range of assumptions are used in practice, with Auer et al 2002 being the closest to this setting. Looking at some of the papers cited, it does however seem overly restrictive. For example, the Bubeck and Cesa-Bianchi paper from 2012 uses much less restrictive assumptions on the reward. I would find it particularly useful for the authors to provide a much more detailed discussion of this point. For example, what if the reward and control variates are only sub-Gaussian? What if the reward has a heavier tail distribution than the control variate or vice versa? I can understand why using Gaussian assumptions is useful for the theory (since it is then simply a linear regression problem with Gaussian variables), but it would be interesting to understand the limitations of this assumption for deriving the theoretical results and whether any of the assumption could be relaxed.
- The numerical experiments section is interesting, but left me wondering how realistic it is to actually use this approach in practice. This is mostly because it did not contain any practical example (as opposed to a synthetic simulation study) where a control variate is actually available. I think the paper would strongly benefit from such an example as otherwise it is difficult to see this actually being a practical method.
- Also, as pointed out in the conclusion, one of the biggest issue will be getting a control variate with known mean. I again struggle to see situations where this would be the case in applied problems. If it is not possible to know the mean exactly, what would be the impact on the estimated reward? Could the bias introduced somewhat lead to sub-optimal results relative to not using a control variate?


Literature review:
-------------------
- There are a number of papers on the use of control variates in reinforcement learning that could be discussed, including:
https://jmlr.csail.mit.edu/papers/volume5/greensmith04a/greensmith04a.pdf
https://arxiv.org/abs/1908.03263
https://arxiv.org/abs/1906.08868
https://openreview.net/pdf?id=H1mCp-ZRZ
Although it is not applied to the same set of RL algorithms, it might still be useful to tell the reader about this for context. Note that I have just listed the first few that come up on Google, but a more complete literature review could help.

- Control variates are also very common in Monte Carlo methods; for example:
https://link.springer.com/article/10.1007/s11222-012-9344-6
https://rss.onlinelibrary.wiley.com/doi/abs/10.1111/rssb.12185
https://projecteuclid.org/journals/bernoulli/volume-25/issue-2/Convergence-rates-for-a-class-of-estimators-based-on-Steins/10.3150/17-BEJ1016.short
These are constructed somewhat differently from the control variates in here, and it might be interesting to discuss the advantages and disadvantages of each approach.

Minor comments
---------------
- line 93: "which could then used as side information"
- Eq 4 and line 161: slightly inconsistent use of * or \star

**Time Spent Reviewing:**

2

---

> ### Author Response · Authors · 2021-08-10
> **Justification joint Gaussian distribution, impact of estimated mean of control variates, experiment with practical exams**
>
> Thank you for the detailed comments and suggestions. We have responded to each of the points below.  All suggestions will be incorporated in the revision.
>
> ### Unclear to me whether assuming that the reward and control variate of each arm has a multivariate normal distribution is something reasonable in practice. What if the reward and control variates are only sub-Gaussian? What if the reward has a heavier tail distribution than the control variate or vice versa?
>
> In the study of multi-armed bandits, the focus is on establishing finite-time bounds, and it is common to assume a specific class of distributions to bring out the main ideas of the algorithm and analysis. Below are some examples of specific classes considered in the literature:
>
> * Bubeck and Cesa-Bianchi consider arbitrary distributions but with bounded support. The bounded support assumption provides a natural bound on the variance, which is then used in the algorithm.
> * The book `Bandit Algorithms' by Lattimore and Szepesvari considers sub-Gaussian distributions but assumes that the sub-Gaussianity parameter is known, which again acts as a proxy for the bound of variance and is used in the algorithms.
> * Auer et al. considered Gaussian distribution to derive the celebrated UCB-Normal algorithm. But to establish its finite-time regret bounds, they use conjectures on the tail behavior of $t$-distribution (see Appendix B) as it is hard to prove them.
>
> Note that classical bandits analysis assumes IID samples. With the application of control theory to bandits, one has to work with non-IID samples, which makes the analysis more challenging. As our work is the first to apply control variate theory to bandits, it is easier to bring out the main ideas using the Gaussian distributions. In fact, the assumption of multivariate normal distribution on rewards and control variates pair is the least restrictive assumption needed for showing finite-time regret guarantees without making any conjectures. Hence, we strongly believe that assuming a multivariate normal distribution is reasonable to show the benefits of using control variates in bandit algorithms.
>
>
> ### Did not contain any practical example (as opposed to a synthetic simulation study)
>
> Our work is a theoretical work that quantifies the gain one obtain by using control variates. It is common in bandit literature to measure the performance of bandits algorithms on synthetically generated data. We have done a detailed validation of our algorithms using several bandits instances.  In the future, we would set up an elaborate experiment to demonstrate the effectiveness of our algorithms on real datasets.
>
>
> ### What would be the impact on the estimated reward? Could the bias introduced somewhat lead to sub-optimal results relative to not using a control variate?
>
> To the best of our knowledge, this is the first work on using control theory to bandits. To establish the gains analytically, we have to assume that mean of the control variates is known. We agree with the reviewer that we may only know the mean of control variate approximately in many applications, and one needs to quantify the degradation in performance. Though the theory of control variates is well established, we found only a few papers that analyze the impact of approximate mean values of control variates, like
>
> * B. W. Schmeiser, M. R. Taaffe, and J. Wang, “Biased control-variate estimation,” IIE Transactions, vol. 33, no. 3, pp. 219–228, 2001.
>
> * R. Pasupathy, B. W. Schmeiser, M. R. Taaffe, and J. Wang, “Control variate estimation using estimated control means,” IIE Transactions, vol. 44, no. 5, pp. 381–385, 2012.
>
> The impact of such approximations in bandits needs to be analyzed extensively and demands independent work. We want to take it as our next task.
>
>
> ### Regarding related work on control variates
> Thank you for suggesting the papers. In the revised version, we will discuss them.

---

### Official Review · Reviewer_bigw · 2021-07-09

**Rating:** 7
**Confidence:** 3

**Summary:**

This paper considers low regret learning in stochastic multi-armed bandits in the presence of control variables. For jointly Gaussian reward and control variables, and linear control variates, the authors show a factor 1-ρ^2 reduction in the regret.

**Limitations And Societal Impact:**


No negative social impact.

One of the shortcomings is that the motivating examples are sketchy and weak. The authors speak about examples from queueing and wireless fading channels, but they do not give concrete examples where such CVs are available, or can be used to improve learning performance.

**Main Review:**

The paper focuses on reducing regret in multiarmed bandits using control variates. The authors derive the regret under linear control variates for the joint Gaussian case, and show that the regret reduces by a multiplicative factor 1-ρ^2 .

For general reward and CV distributions, the authors utilise jackknifing and splitting resampling to obtain estimators with smaller bias and sharper confidence intervals. For the general case, the confidence intervals are only asymptotic and hold approximately for finite samples.

Overall, the paper is reasonably well written, and seems technically sound. It takes a well-known technique of CVs and applies it in the context of MABs using standard techniques. The innovativeness of the work can be characterised as “moderate” in my opinion.

Minor:

(I) Line 165: Please clarify why \bar{X}_i are correlated
(II) Line 176: It is not clear how lemma 2 is analogous to Hoeffding from the way you have written it in terms of the t-distribution. Please elucidate by comparing to the exponential decay in Hoeffding
(III) Line 322: “our the bounds”

**Time Spent Reviewing:**

4

---

> ### Author Response · Authors · 2021-08-10
> **Novelty of the work and motivating examples**
>
> Thank you for the detailed comments and suggestions. We have responded to each of the points below.  All suggestions will be incorporated in the revision.
>
> ### Innovativeness of the work
>
> Application of linear control variate to bandits to variance reduction is new. To make the control variate theory work for bandits, one needs to carefully construct confidence intervals using the correlated samples, which makes the regret analysis challenging. We handled it by carefully bounding the expected number of pulls of the sub-optimal arms. Note that the UCB-indices in our algorithms are based on the estimate of the variance of the sample reward estimator and not on the estimate of the variance of rewards (as in UCBV). We believe adaptation of control variate to bandit theory required significant work, and we have achieved it in this work.
>
>
> ### Why $\bar{X}_i$ are correlated
> After observing reward and associated control variates for arm $i$, the value of $\hat\beta_i^\star$ is re-estimated. The value of $\hat\beta_i^\star$ depends on all observed rewards and associated control variates from arm $i$. Since all $\bar{X}\_{., i}$ uses same $\hat\beta_i^\star$, it leads to correlation between the $\bar{X}_{., i}$ observations.
>
>
> ### How Lemma 2 is analogous to Hoeffding from the way you have written it in terms of the t-distribution
> We are comparing our bound with the UCB1 algorithm. We also have similar terms in UCB1 analysis where confidence term $V_{t,s,1}^{(\alpha)}\sqrt{{\hat\nu_{s,i}}}$ in Lemma 2 is $\sqrt{\frac{\alpha \log t}{s}}$. When we use Hoeffding inequality in UCB1, we get $\mathbb{P}$ { $|\hat\mu_{s,i} - \mu_i| \ge \sqrt{\frac{\alpha \log t}{s}}$ } $\le \frac{2}{t^{2\alpha}}$, where $\hat\mu_{s,i}$ is the empirical mean of arm $i$ using $s$ observations. The probability of the estimated mean deviating from the true mean of arm $i$ decays fast in the both cases ($\frac{2}{t^{\alpha}}$ for Lemma 2 which uses t-distribution based confidence intervals and $\frac{2}{t^{2\alpha}}$ for UCB1 which uses Hoeffding Inequality).
>
>
> ### Motivating examples are sketchy and weak.
> All our motivating examples are valid where control variates are available. However, we acknowledge that the mean of the control variates may not be known with full accuracy. In many applications, control variates can be constructed such that their mean value is known (see Eq. (7) and Eq. (8) of Kreutzer et al., 2017).
>
> This work largely focused on how to leverage the control variate theory to improve the performance of bandit algorithms. As a next step, we aim to study how the regret performance gets affected when the mean of the control variates are known only approximately.

---

> > ### Comment · Reviewer_bigw · 2021-09-11
> > **After author response**
> >
> > Thanks for your detailed responses. I am happy to increase the score towards acceptance (7).

---

> > > ### Author Response · Authors · 2021-09-11
> > > **Please update the paper's rating**
> > >
> > > Dear bigw,
> > >
> > > We thank you for increasing the score (rating). We kindly request you to please update the paper's official rating as well.
> > >
> > > Thanks and best regards,
> > >
> > > Authors

---

### Official Review · Reviewer_7d9W · 2021-07-15

**Rating:** 7
**Confidence:** 3

**Summary:**

The paper tackles a bandit setting similar to contextual bandits, where the feedback consists of the reward obtained from playing an arm, but also an additional random variable (the control variate) which is correlated with the reward but whose mean is known to the decision maker. Unlike in contextual bandits, the value of this control variate is not available at decision time but it is only revealed after the arm's play and can thus only be used to produce better estimates of the rewards of arms. The paper proposes the UCB-CV algorithm that builds confidence intervals that exploit this additional information to produce lower regret. The algorithm is accompanied by a regret analysis and experiments demonstrating the algorithm's performance and the decrease in regret that can be achieved based on the correlation coefficient between the reward and the control variates.

**Limitations And Societal Impact:**

The paper does a decent job addressing limitations of their work in the 3 remarks at the end of Section 4, though this might not be comprehensive.

**Main Review:**

Overall I found the paper well written and presented. I appreciate the effort put into the experimental section as well as the Section 2 offering introduction to the area. My main concerns surround the practical relevance of the setting (and algorithm - relative to Splitting for instance), the unclear distinction of the results here from prior work (a fence between the problem tackled here and contextual bandits, and MABs with covariates). Detailed review below:

**Originality**:\
To the best of my knowledge this setting is original for the bandit setting. I do however have a few questions regarding the practical relevance of the problem studied here. First, the control variates would be available at decision time in most practical scenarios (the job "size" is usually known ahead of time, similarly wouldn't the pilot signals be sent in advance of the packet?). Secondly, from the historic data used to estimate the means of the control variates, we can also estimate the variance. What is the reason why the variance of the control variate ${\sigma^2_{w,i}}_{i \in [K]}$ is left unknown?\
I think there is a bit of room for improvement when it comes to separating the setting here from contextual bandits. If the control variates were revealed at the begining of the round (like contexts are revealed in contextual bandits), would this setting be a specific instance of contextual bandits?\
I am also missing how this setting compares to the MAB problem with covariates (see [1], [2]).

**Quality**:\
I have not checked the proofs in the Appendix. The paper seems technically sound and the algorithm well-grounded in theory. The authors also point out several remarks offering insights into the bound presented in Theorem 1. I would have liked to see a bit more detail as to where the upper bound on $(V^{(2)}_{T,T,q})^2$ is obtained. Possibly also include larger time horizon T in the plot (order of thousands or tens of thousands).\
I appreciate the effort put into the experiments section and implementing a good number of baselines. I would like to know more about why Splitting outperforms all algorithms and why would UCB-CV be preferable?\
A further question surrounds what is the impact of the hyperparameter $\alpha$ that is fixed to $\alpha=2$ throughout the majority of the paper. Is this the optimal value for this parameter? If not, how sensitive is the algorithm to the choice of $\alpha$?

**Clarity**:\
I found the paper well presented with sufficient attention given to aspects that cause confusion (I particularly appreciate the clarifying remarks following Theorem 1.) I would appreciate more clarifications surrounding the novelty of the setting and the advantages of the proposed method relative to just using Splitting which appears to outperform UCB-CV in the experiments (an example of a problem instance where Splitting would fail would present a convincing argument).

**Significance**:\
The significance of the results should be stated more clearly in the paper. As is, I believe the setting studied here has limited connection to practical applications, though I do see the potential of future study into the theory of exploiting control variates (I liked the ranking example in Section 7).


**Time Spent Reviewing:**

6 hours

---

> ### Author Response · Authors · 2021-08-10
> **Distinction from contextual bandits, performance of Splitting algorithm, connection to practical applications**
>
> Thank you for the detailed comments and suggestions. We have responded to each of the points below.  All suggestions will be incorporated in the revision.
>
> ### Novelty of the setting
>
> The novelty of the work is to adapt the linear control variate theory to bandit algorithms and characterize the gain. To the best of our knowledge, this is not done before. Several works have studied how to exploit side-information in contextual settings. However, when side-information is available in the form of a control variate, there are no works on how it can be exploited to improve the performance of the bandit algorithm. Our work fills this gap. Moreover, the analysis of bandit algorithms with control variates is not easy as reward samples are no longer independent. Our work addresses these challenges by appropriately constructing the confidence bounds.
>
> ### Distinction of the results here from prior work (contextual bandits and MABs with covariates)
>
> Control variates cannot be treated in the same way as contextual information or covariates. The contextual information, covariates, or control variates are all some form of side-information. When contextual and covariates are used, it is assumed that they are available in each round before an action is taken. Whereas control variates need not be available in each arm before an arm is played. Hence, we cannot use the result on contextual bandits or MABs with covariates to study bandits with control variates.
>
> We note that control variates and covariates sound strikingly similar through the names, but they are very different! In fact, covariates and contextual information are the same -- it is just that when contextual information is used, the reward is assumed to some parametric function (like linear, GLM),  whereas only smoothness assumption is made when using covariates.
>
> In summary, the following are the main difference between multi-armed bandits with control variates (MAB-CV) and the popularly  Linear Contextual Bandits (LCBs).
>
> 1. LCB are structured bandits where the reward is assumed to be a linear function of context/side-information, and the mean reward varies with side-information. In contrast, we consider an unstructured setup where no explicit relation is assumed between mean rewards and side-information (control variates). Unlike in LCB, mean rewards do not vary with side-information in MAB-CV.
> 2. In LCB, it is assumed that the side-information is available for each arm before an arm is played. The side-information (control variates) in our setup may not be available before an arm is played. For example, you can use a pilot packet to measure channel gain only when that channel is selected.
>
> ### What is the reason why the variance of the control variate $\sigma^2$ is left unknown?
>
> When the variance of control variates is known, it can be directly used to estimate $\beta$ (replacing estimated variance by variance in the denominator of Eqs. 5). Our work is more general as it with lesser assumptions. When variance information is available we can only do better.
>
> ### Possibly also include larger time horizon T in the plot (order of thousands or tens of thousands) for the upper bound on $V_{T, T, q}^{(2)}$
>
> The plot is already in the paper. Please check Figure 1(b). Note the $x$-axis has a range of $1e^8$.
>
> ### I would like to know more about why Splitting outperforms all algorithms and why would UCB-CV be preferable?
>
> Note that the performance guarantee of UCB-CV holds under the assumption that reward and control variate are jointly Gaussian. When this assumption is violated, it is hard to provide a finite-time guarantee for the UCB-CV algorithm. Hence we developed the Splitting algorithm extending the ideas from jointly Gaussian distribution to General distributions. The Splitting algorithm is a heuristic algorithm based on the re-sampling method, and it comes with no finite time guarantees. Splitting and UCB-CV perform almost identically on jointly Gaussian distribution. However, when applied to general distributions, the UCB-CV did not do better than splitting as it is optimized only for Gaussian distribution. The better performance of Splitting in general distribution can be attributed to the re-sampling method, which will have more degrees of freedom in calculating the UCB-indices.
>
> ### What is the impact of the hyper-parameter $\alpha$ that is fixed to $\alpha=2$ throughout the majority of the paper. Is this the optimal value for this parameter? If not, how sensitive is the algorithm to the choice of $\alpha$?
>
> $\alpha$ is a hyper-parameter to trade-off between exploration and exploitation. Our algorithm works as long as $\alpha>1$ and it effect the regret bound through the term $\sum_{i=1}^\infty \frac{1}{t^\alpha}$. This term has a nice closed form value of $\pi^2/6$ when $\alpha=2$. Hence we state the results with $\alpha=2$ and use it in the experiments as well. If we keep increasing $\alpha$ above $1$, it is clear that regret performance only deteriorates.
>
> ### If the control variates were revealed at the beginning of the round (like contexts are revealed in contextual bandits), would this setting be a specific instance of contextual bandits?
>
> Yes. This problem can be modeled as contextual bandits when the control variates for all arms are revealed to the learner before arm selection. Let $W_t$ be the control variates revealed to learner in round $t$, $Y_{t,i} = (1, W_t)$, $\theta_i = (\mu_{i}, 1)$, and $\epsilon \sim \mathcal{N}(0, \sigma_{i}^2)$. Then, the reward observed by learner for arm $i$ is given by $X_{t,i} = Y_{t,i}^\top \theta_i + \epsilon$. This is similar to the formulation given by Reviewer 1. However, there are two issues here.
>
> 1. For this to work, $E[W_i]=0$ for all $i$ and
>
> 2. The values of $W_i$ need to be known for all $i$. Both of which may not hold and we cannot apply the contextual bandits directly.
>
>
>
> ### Limited connection to practical applications
>
> We have given two motivating examples (Job Scheduler and Communication network) in the paper. There are several other applications where our work finds application. Consider the problem of showing top restaurants to a user by the online food delivery platform. The restaurants are rated by the users, but apart from users' rating, the online platform also collects auxiliary information (delivery time, time taken for food preparation, etc.) when a user orders from a restaurant.  Note that the restaurant's rating given also depends on overall delivery time. Here, the collected auxiliary information can be used as control variates. Now consider the problem of best cab recommendation to the rider by online cab aggregator (e.g., Uber). Each cab is rated by the rider, but apart from the rider's rating, the online cab aggregator collects auxiliary information (cab's distance from the rider, driver response to ride request, etc.) that influences the rider's rating. Such auxiliary information can be used as control variates. Similar situations also appear in e-commerce platforms while recommending sellers to buyers. Here auxiliary information can be the seller's response time for order confirmation and delivery, quality of item packing, etc., which affects the sellers' rating given by users.

---

### Official Review · Reviewer_trp8 · 2021-07-16

**Rating:** 7
**Confidence:** 4

**Summary:**

Starting from the hypothesis that smaller variance estimators of mean rewards result in better bandit performance, the authors leverage Control Variate (CV) theory to propose a bandit algorithm under the presence of auxiliary information about arm rewards.

The authors provide theoretical regret bounds of the proposed control variate based UCB algorithm (under the multivariate normal distribution assumption) and experimentally validate the performance on Gaussian and other distribution based assumptions.

The authors show that, as expected, the attained regret reduction is a function of how strongly correlated the reward samples and the observed control variates are.

**Limitations And Societal Impact:**

The authors discuss in their conclusion section the fact that the proposed algorithm assumes knowledge of the mean of the control variates per-arm.

The analysis (at least empirically) of the impact of estimating these control variate means on regret performance would significantly strengthen their work.

**Main Review:**

The contribution of this work is on the combination of Control Variate theory (well-known in the statistics community) with a state of the art bandit algorithm (UCB), along with its corresponding regret bounds. The experimental evaluation section however can be significantly improved.

The analysis of the confidence intervals is challenging and novel in the context of the bandit literature, because the optimal (linear) control variate parameters need to be estimated from the observed reward samples (as determined by the bandit algorithm). The authors present in Section 4 the regret bound of the proposed algorithm for multivariate Gaussian rewards and leave for Section 5 the adaption of the algorithm (via jackknifing, splitting, and batching methods) for general distributions.

Lemma 1 and Lemma 2, where control variate theory is used to design confidence intervals of the proposed control variate mean reward estimators, are key for their regret bound in Theorem 1, where the regret reduction induced by control variates is reflected via the correlation parameter $\rho$. The authors provide a very informative discussion in Remarks 1-3 and Figure 1 on the non-trivial dependency between number of covariates $q$ and the constants appearing in Theorem 1.

The proposed algorithm assumes knowledge of the mean $w$ of the control variates per-arm (a limitation only discussed in their conclusion).
- The paper could be significantly improved if a theoretical analysis of the error introduced by estimates of the control variate mean $\widehat{w}$ were to be provided.

In order to extend their control variate based approach to non-Gaussian distributions (where the properties of t-distributions can not be used to obtain analytical confidence intervals), the authors resort to well-known resampling methods to develop (approximate and/or asymptotic) confidence intervals.

Finally, to illustrate the performance of the proposed algorithm and the validity of the presented regret bound, synthetic evaluations are presented in Section 6. However, the results would be more convincing if the following suggestions were to be incorporated:

1. The proposed algorithm assumes knowledge of the mean of the control variates $w$, which is an unreasonable assumption in practice. As such, it would be informative to show, empirically, how much the performance of the proposed algorithm degrades with the quality of the control variate mean estimates $\widehat{w}$.

2. The studied bandit instances can be posed as linear contextual bandits:
   - $y=x^\top \theta + \epsilon$, with $y_{t}=X_{t}$, $x_{t}=(1, W_{t})$ and $\theta=(\mu_{v}, 1)$, where $\theta$ are the unknown parameters, $x_t$ the observed context (with a bias term) and $\epsilon \sim N(0, \sigma_{v}^2)$.
   - A set of experiments where their proposed algorithm is compared to linear Gaussian Thompson sampling and UCB algorithms would be very insightful to better illustrate the benefits of control variates in the linear-Gaussian setting.

3. The authors state that "We observe that Thomson Sampling has a large regret". Given the successful performance presented by many in the literature with Thompson sampling for Gaussian bandit rewards, I encourage the authors to clarify what mean and variance priors they used for their Gaussian Thompson sampling implementation, and whether adjusting them improves the observed performance.

Other minor comments/suggestions:

- Line 15: "the arms rewards" -> "the arm rewards"
- Line 165: "CVs have follow multivariate" -> "CVs follow a multivariate"
- Line 276: "regret bound with" -> "and regret results with"?


**Time Spent Reviewing:**

2

---

> ### Author Response · Authors · 2021-08-10
> **Connection with Contextual Bandits, Comparison with Thompson Sampling, and effect of using estimated mean of control variates**
>
> Thank you for the detailed comments and suggestions. We have responded to each of the points below.  All suggestions will be incorporated in the revision.
>
> ### The studied bandit instances can be posed as linear contextual bandits:
>
> MAB with Control Variates (MAB-CV) cannot be posed as instances of Linear Contextual Bandits (LCB) for the following reasons.
>
> 1. Note that to use the model $y=x^T\theta+\epsilon$, with $y_t=X_t, x_t=(1,W_t)$ and $\theta=(\mu_v, 1)$, where $\theta$ are unknown parameters and $x_t$ is the context, we need to know the values of $x_t$ for all the arms, i.e., side information $W_t$ to be known for all the arms. However, it is not necessary that $W_t$ are known for all the arms beforehand.  In fact, in many applications (like channel selection), $W_t$ for an arm is only known when that arm is selected.
>
> 2. Even if $W_t$ are known for all arms, the setup in (1) will work provided $\mathbb{E}[W_t]=0$ for all arms and $t$, which is very restrictive to some. However, a simple workaround is to redefine parameter $\theta$ as $\theta=(\mu_v-\omega_v, 1)$, where $\omega_v =\mathbb{E}[W_t]$.
>
> 3. LCB are structured bandits where the reward is assumed to be a linear function of context/side-information, and the mean reward varies with side-information. In contrast, we consider an unstructured setup where no explicit relation is assumed between mean rewards and side-information (control variate). Unlike in LCB,  mean rewards do not vary with side-information in MAB-CV
>
> 4. In LCB, it is assumed that side-information is available for each arm before an arm is played. The side-information (control variate) in our setup may not be available before an arm is played.
>
>
> ### Regarding Thompson Sampling
>
> As the arm rewards are normally distributed in the problem instance $1$ and $2$, we have implemented 'Algorithm 2: Thompson Sampling using Gaussian priors' of Agrawal and Goyal, 2013. We ran our initial experiments on two easy problem instances where the sub-optimality gap is very large ($10$ times larger than Instance $1$ and $2$ considered in the paper). These problem instances are as follows: In the first problem instance, we set $\mu_{v,i} = 6 - (i-1)*0.5$ and $\mu_{w,i} = 3$ for arm $i \in [K]$. The value of $\sigma_{v,i}^2 = 1$ and $\sigma_{w,i}^2 = 1$ for all arms. The second problem instance is the same as the first, except the mean value of the control variates are set as $\mu_{w, i} = 8 - (i-1)*0.5$ for all $i$. We observed that Thompson Sampling has almost linear regret for easy problems (which we believe is due to the high variance of arms rewards that leads to overlapping of rewards distribution's space). Therefore, we ignore Thompson Sampling for more hard problems (Instances $1$ and $2$ in the paper, which is having a smaller sub-optimality gap). We reran the experiment on the problem instances described in the paper and observed that Thompson Sampling does not have linear regret (due to smaller variance) but still performing poorly as compared to UCB-CV. The results are shown in the following table:
>
> | Algorithm/Regret  | Instance 1      | Instance 2      | Easy Instance 1      | Easy Instance 2     |
> |-------------------|-----------------|-----------------|----------------------|---------------------|
> | UCB1              | 832.004 ± 3.087 | 499.244 ± 1.908 | 1859.4 ± 12.861      | 1269.33 ± 10.226    |
> | UCBV              | 412.150 ± 1.685 | 386.333 ± 0.981 | 1959.615 ± 12.464    | 1874.43 ± 6.966     |
> | Thompson Sampling | 444.782 ± 2.465 | 278.237 ± 1.567 | 11000.225 ± 1211.982 | 23200.08 ± 2422.635 |
> | EUCBV             | 204.491 ± 1.890 | 111.170 ± 1.001 | 836.2 ± 10.220       | 501.87 ± 7.719      |
> | UCB-CV            | 53.312 ± 0.596  | 64.995 ± 0.413  | 398.08 ± 4.400       | 847.17 ± 19.877     |
>
>
> We have reported the average cumulative regret over $100$ runs for $50000$ rounds for different algorithms on Instance $1$ and $2$, which is the same as given in the paper. Whereas we have reported the average cumulative regret over $100$ runs for $5000$ rounds (as the difference is too significant) for different algorithms on Easy Instance $1$ and $2$ as defined in the above paragraph. We have added a $95$% confidence interval into the average cumulative regret. We observe that the cumulative regret of Thompson Sampling is very large on easy problem instances (see 4th and 5th columns of Table). However, Thompson Sampling performs well on hard problem instances but still poorly performing than UCB-CV by significant margins as shown in the 2nd and 3rd columns of Table.
>
>
> ### The proposed algorithm assumes knowledge of the mean of the control variates (CV) $(\omega)$, which is an unreasonable assumption in practice. How much the performance of the proposed algorithm degrades with the quality of the control variates mean estimates $\hat\omega$?
>
> We agree that mean of control variate may not be available in many applications. However, it is not unreasonable as the control variates can be constructed such that its mean value is known (see Eq. (7) and Eq. (8) of Kreutzer et al., 2017). If the mean value is unknown, we estimate it from the samples, but as rightly pointed out, using the estimated/approximate means can deteriorate the performance.
>
> Though the theory of control variate is well developed for known mean, only empirical studies are available to the case when the means of control variates are known approximately. To the best of our knowledge, the following two papers analyze the impact using approximate means.
>
> * B. W. Schmeiser, M. R. Taaffe, and J. Wang, “Biased control-variate estimation,” IIE Transactions, vol. 33, no. 3, pp. 219–228, 2001.
>
> * R. Pasupathy, B. W. Schmeiser, M. R. Taaffe, and J. Wang, “Control variate estimation using estimated control means,” IIE Transactions, vol. 44, no. 5, pp. 381–385, 2012.
>
> Our work has demonstrated the applicability of control variates to bandit theory and is an important first step. We agree that the impact of knowing only the approximate mean of control variates needs to be systematically analyzed in the bandit setting and can be an independent work itself as the experiments have to be exhaustive. We aim to take it as the next work.
>
> To know the effect of approximation error $(\epsilon)$ in the mean estimation of control variates, we run an experiment on Instance $5$ given in the paper. We assume that the approximated mean of control variates is given by $\omega_i + \epsilon$. We vary the value of $\epsilon$ and report the average cumulative regret over $100$ runs with a $95$% confidence interval in the following table.
>
> | Algorithm | Cumulative Regret for Instance 5|
> |-----------------------|-----------------------------------------------------------------|
> | UCB-CV (with $\epsilon=0$)  | 435.11 ± 4.434      |
> | UCB-CV (with $\epsilon=-1$) | 704.565 ± 12.251    |
> | UCB-CV (with $\epsilon=1$)  | 694.33 ± 8.187      |
> | UCB-CV (with $\epsilon=-2$) | 1437.795 ± 19.661   |
> | UCB-CV (with $\epsilon=2$)  | 1443.125 ± 15.371   |
> | UCB1                        | 2342.31 ± 16.59     |
> | UCBV                        | 2187.29 ± 9.77      |
> | Thompson Sampling           | 22250.19 ± 2349.224 |
> | EUCBV                       | 1029.99 ± 14.496    |
>
>
> We observe that the cumulative regret of UCB-CV increases with an increase in approximation error. UCB-CV can even start performing poorly than the existing state-of-the-art algorithm (EUCBV) for significant large approximation errors.

---

> > ### Comment · Reviewer_trp8 · 2021-08-31
> > **Thank you for your informative responses**
> >
> > Dear authors,
> >
> > I found the explanation on the differences between the Control-Variate and Contextual bandit settings very clear. Similarly, the discussion on the impact of your assumptions (e.g., unknown mean/variances, Gaussian distributions) is very informative. I would encourage to incorporate these insights into the updated manuscript.
> >
> > More importantly, I appreciate your effort in providing new empirical results for both Thompson sampling results and a new study that incorporates mean control variate errors. These results on estimating the mean of control variates, as well as the accompanying discussion, is very informative and improves the overall quality of their work.
> >
> > I have increased my score accordingly.

---

> > > ### Author Response · Authors · 2021-08-31
> > > **Incorporating the changes**
> > >
> > > Thank you. We will definitely incorporate all the insights and discussions in the updated version.

---

### Decision · Program_Chairs · 2021-09-27

**Decision:**

Accept (Poster)

**Comment:**

The committee has appreciated the involved and detailed response to their questions and comments and agreed that this work should be accepted. One reviewer raised the question of the knowledge of the variance of the control variate $\sigma_{w,i}^2$ and would encourage the authors to investigate and discuss this question in their final version.